



# Simultaneous versus sequential estimation of biogeochemical and physical parameters in coupled marine ecosystem models

Skyler Kern[1], Mary E. McGuinn[2], Katherine M. Smith[3], Nadia Pinardi[4], Kyle E. Niemeyer[5], Nicole S. Lovenduski[6], and Peter E. Hamlington[2]

[1]Mechanical Engineering Department, University of Alaska, Anchorage, AK, USA
[2]Paul M. Rady Department of Mechanical Engineering, University of Colorado, Boulder, CO, USA
[3]Los Alamos National Laboratory, Los Alamos, NM, USA
[4]Department of Physics and Astronomy, University of Bologna, Bologna, IT
[5]School of Mechanical, Industrial, and Manufacturing Engineering, Oregon State University, Corvallis, OR, USA
[6]Department of Atmospheric and Oceanic Sciences and Institute of Arctic and Alpine Research, University of Colorado, Boulder, CO, USA

**Correspondence:** Skyler Kern (sjkern@alaska.edu)

**Abstract.** As computational resources have increased in availability and capability, so has the complexity of the models used to represent biogeochemical (BGC) processes in ocean simulations. To effectively calibrate the increasingly large number of uncertain parameters in these models, efficient parameter estimation methods are needed to ensure that the models can accurately represent the BGC processes under investigation. In this study, we address this challenge using a multistage automatic

parameter estimation methodology that sequentially applies global sampling and local optimization to calibrate both the BGC model parameters and the parameters associated with the mathematical representation of physical ocean dynamics. We quantitatively compare the accuracy of sequential and simultaneous parameter estimations of moderately complex BGC and physical models at locations corresponding to the Bermuda Atlantic time series and the Hawaii Ocean time series. The results show that the best overall agreement with the observational data is obtained when the BGC and physical model parameters are estimated

simultaneously, rather than sequentially. In particular, simultaneous estimation results in significantly improved predictions of oxygen and particulate organic nitrogen. Moreover, the agreement is improved in general when the physical model is included in the estimation, as opposed to calibrating the BGC model alone. This study also serves as a demonstration of a meta-algorithm for performing parameter estimation in high-dimensional models with local optimization approaches.

## 1 Introduction

Computational simulations of the ocean on local, regional, and global scales typically rely on physical and biogeochemical (BGC) models whose parameters have been tuned using available observational data (e.g., temperature, current velocity, biogeochemical tracer concentrations). The physical and biogeochemical components of these coupled models, which represent complex physical and BGC processes across a range of spatiotemporal scales (Fox-Kemper and Ferrari, 2008; Henson et al., 2015; Smith et al., 2016; Ruiz et al., 2019), are often tuned sequentially, rather than simultaneously. As such, a tuning exer-

cise for the second model component may result in parameter values that compensate for errors introduced by the first model





component. This type of sequential parameter tuning in coupled models may not produce accurate predictions of the coupled physical and BGC state under new or different ocean conditions, thereby reducing the general applicability and utility of the tuned model parameters.

As a potential solution to this issue, *simultaneous* and *fully coupled* parameter estimations of physical and BGC model components may enable significant improvements in simulation accuracy and generality. In this paper, we quantify the potential benefits of such an approach by comparing sequential and simultaneous calibrations of a moderately complex, coupled physical-BGC model at two different ocean locations. The BGC model is the 17-state variable version of the Biogeochemical Flux Model (BFM17, Smith et al., 2021) and the physical model is the one-dimensional (1D) formulation of the Princeton Ocean Model (POM1D, Blumberg and Mellor, 1987). This combined biophysical model (termed BFM17+POM1D) was initially developed by Smith et al. (2021) from the 56-state variable BFM (Vichi et al., 2007) with parameter values that were manually adjusted to give good agreement with data for open ocean oligotrophic BGC conditions from the Bermuda Atlantic time series (BATS, Steinberg et al., 2001). Kern et al. (2024) further improved the accuracy and generality of BFM17 by developing and applying an efficient automated parameter estimation approach based on a three-step optimization method for BGC models with high-dimensional parameter spaces. Using the resulting optimized model, Kern et al. (2024) demonstrated good agreement between BFM17+POM1D and observational data from both BATS and the Hawaii Ocean time series (HOTS, Karl and Lukas, 1996).

Here, we employ the same three-step optimization method developed by Kern et al. (2024) to examine the relative benefits of sequential versus simultaneous calibrations of the BGC and physical model components. The method treats parameter estimation as a constrained optimization problem using a global search followed by local gradient-based optimization. The use of local gradient-based optimization efficiently identifies an optimal solution, while the global search is used to avoid biasing the solution with *ad hoc* initial guesses. The best case among the optimized parameter sets is then selected as the optimal solution. Although this method resulted in a formulation of BFM17+POM1D that agreed well with the data from BATS and HOTS, Kern et al. (2024) considered only the estimation of BGC model parameters, introducing the possibility that the BGC model was tuned to compensate for inaccuracies in the parameterization of physical processes. In the present study, we examine this issue by estimating 42 total parameters in the coupled biophysical model BFM17+POM1D, comprising the 12 most sensitive parameters of the BGC model, 6 physical boundary condition and turbulence closure parameters, and 24 bimonthly vertical eddy mixing parameterization coefficients (also representing physical processes). We then quantitatively compare the accuracy of sequential versus simultaneous estimations of the BGC and physical model parameters at BATS and HOTS.

The paper is organized as follows. Section 2 provides a brief description of the parameter estimation methodology. The biophysical model BFM17+POM1D is then described in Section 3, with the two implementations and the baseline results following in Section 4. In Section 5 we discuss how we selected the BGC parameters included in the parameter estimation studies, while Section 6 presents the results of twin simulation experiments with these parameters. Section 7 presents the results of the two calibration cases for both model implementations. Finally, the main conclusions from this study are presented in Section 8.





## 2 Optimization Methodology

The parameter estimation methodology used in this paper was originally described in Kern et al. (2024) and treats parameter estimation as a constrained optimization problem, where a measure of the difference between observational data and model output fields is minimized. The problem is formulated as

$$\min_{\mathbf{c}} \mathcal{J}(\mathbf{c})$$
$$\text{subject to } \mathbf{c}_{\min} \leq \mathbf{c} \leq \mathbf{c}_{\max}, \tag{1}$$

where $\mathcal{J}(\mathbf{c})$ is the objective function evaluated for a set of parameter values expressed by the vector $\mathbf{c}$. The parameters are bounded by the set of minimum and maximum values expressed by the vectors $\mathbf{c}_{\min}$ and $\mathbf{c}_{\max}$, respectively. For the parameter estimation studies in Kern et al. (2024) and here, the objective function is an error metric that quantifies the disagreement between the model results and the observational data.

The parameter estimation method was developed to incorporate data for multiple objective fields from multiple ocean locations. The most general expression of the objective function is therefore

$$\mathcal{J}(\mathbf{c}) = \sum_{i=1}^{N_{\mathrm{s}}} \sum_{j=1}^{N_{\mathrm{v}}} \Pi_{ij} \delta_{ij}(\mathbf{c}), \tag{2}$$

where $N_{\mathrm{s}}$ is the number of target sites, $N_{\mathrm{v}}$ is the number of target fields, $\Pi_{ij}$ is a weighting factor, and $\delta_{ij}(\mathbf{c})$ is a measure of the disagreement between model output and observational data. In all subsequent equations, $i$ will index location, while $j$ corresponds to the target field. The parameter estimation cases performed here quantify the difference between model and observational data using the normalized root-mean-squared difference (RMSD), expressed as

$$\delta_{ij}(\mathbf{c}) = \frac{1}{\sigma_{ij}^{(\mathrm{obs})}} \left\{ \overline{\left[ V_{ij}^{(\mathrm{obs})}(\mathbf{x}, t) - V_{ij}(\mathbf{x}, t; \mathbf{c}) \right]^2} \right\}^{1/2}. \tag{3}$$

Data are compared as monthly averaged vertical profiles where $V_{ij}(\mathbf{x}, t; \mathbf{c})$ is the model output field for a given parameter set and $V_{ij}^{(\mathrm{obs})}(\mathbf{x}, t)$ is the corresponding target observational field. The RMSD is normalized by the standard deviation in the observational field in all months and at all depths, $\sigma_{ij}^{(\mathrm{obs})}$.

Equation (2) includes weights, $\Pi_{ij}$, for each field of interest. These weights can be used to modulate the relative importance of the different fields during parameter estimation, but here we set them equal to 1. This choice causes the optimization to only factor in the relative magnitudes of the normalized RMSD values, without any *ad hoc* decision on the relative reliability or importance of the observational data sets. Normalization with the standard deviation is primarily included to non-dimensionalize the RMSD values so they can be summed into a single objective value. In this study, we explore the sequential versus simultaneous optimization of physical model parameters separately at the BATS and HOTS sites, such that $N_{\mathrm{s}} = 1$ for all optimizations.

The method developed in Kern et al. (2024) seeks to efficiently optimize a large set of BGC model parameters. This goal was achieved by balancing the strengths and weaknesses of global and local optimization techniques. Global optimization techniques will generally uncover the global minimum, ensuring the best possible solution. However, these techniques are





computationally expensive, which limits the number of parameters that can be optimized simultaneously. Local methods –
especially those that are gradient-based – are much more reasonable in terms of the number of model evaluations performed,
even for high-dimensional parameter spaces. However, this class of optimization algorithms falls victim to poor initial guesses
and local minima.

The methodology applied here uses multiple local optimizations from initial points selected from the best of those included
in a global search. This allows us to avoid poorly selected initial parameters for the local optimizations, while also avoiding
limiting the optimization to a small region of the overall parameter space. Due to the unrealistic number of runs required to
fully search the parameter space, the initial search is truncated based on the available computational resources. Although this
approach still does not guarantee a global minimum, it has the advantage of reducing the impact of arbitrary user decisions on
optimization results.

Overall, the methodology consists of three primary steps:

1. The parameter space is probed $N_{\mathrm{samp}}$ times with a global search or optimization technique;

2. The $N_{\mathrm{top}}$ parameter sets are selected and used to initialize local optimization runs;

3. The results of the local optimizations are compared to select the final calibrated parameter set.

The methodology is very flexible in its most general form, leaving room for the selection of different sampling and optimization
methods, the number of parameters sampled, and the number of local optimization runs.

The sampling and optimization are performed here with the coupled biophysical model using DAKOTA (Adams et al., 2019).
DAKOTA is an open-source numerical toolbox that allows various sampling, optimization, and uncertainty quantification
algorithms to be applied to an arbitrary model. Figure 1 is a schematic of BFM17+POM1D, which will be discussed in the next
section, and its coupling to DAKOTA. DAKOTA is able to apply a variety of algorithms to an arbitrary model by treating the
model as a "black-box," as indicated by the dashed line in Fig. 1. DAKOTA is not integrated with the model and instead interacts
with the model through an interface program. DAKOTA outputs a set of parameters to be tested, runs the interface program,
and then reads the value of the resulting objective function. The interface interprets the parameters, runs the simulation, and
runs the objective function calculator.

The parameter estimations performed here use the standard Latin Hypercube Sampling algorithm from DAKOTA for Step
1. The local optimizations (Step 2) are performed using a Quasi-Newton (QN) algorithm provided by the Opt++ library within
DAKOTA (Meza et al., 2007). Opt++ is a C++ class library that uses object-oriented programming for nonlinear optimization
algorithms. The object-oriented toolbox provides a relatively simple way to select different modules that set the optimizer and
its behavior in a manner consistent with the larger DAKOTA framework. The QN method was used in Kern et al. (2024) after
testing various algorithms available in DAKOTA. It performed the best by reliably and efficiently converging to a solution.
Other algorithms, especially conjugate gradient, were not able to converge either efficiently or at all due to the topography of
the objective space.





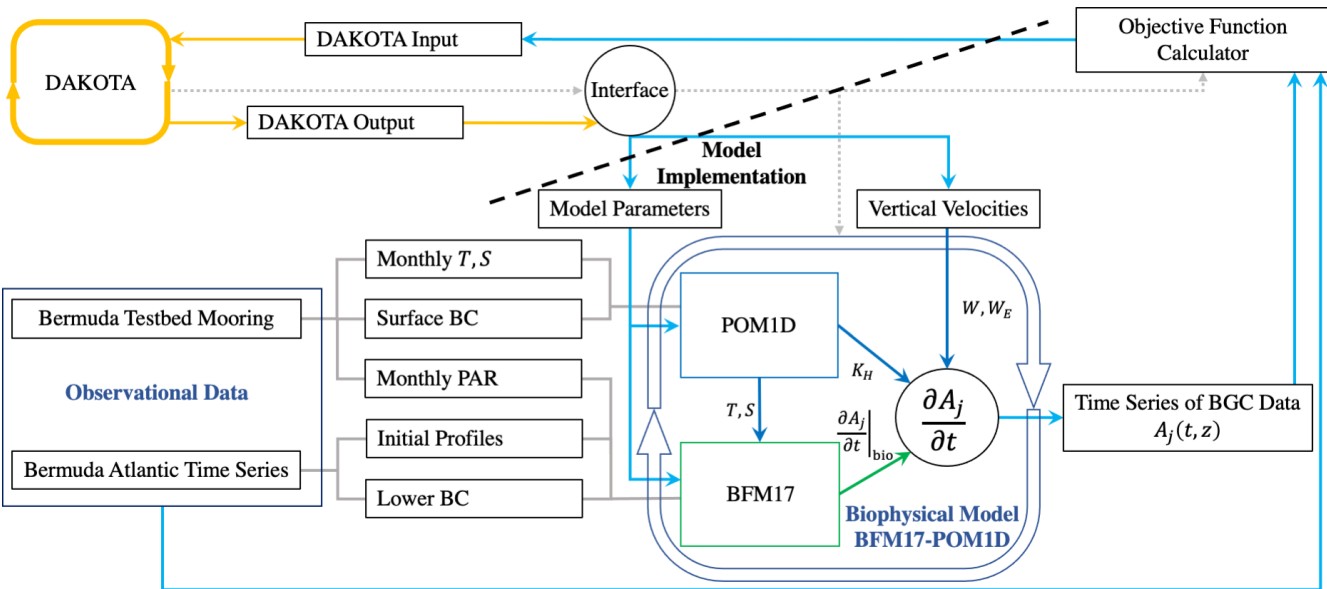

**Figure 1.** Schematic showing how DAKOTA interfaces with BFM17+POM1D. The schematic also shows how BFM17 and POM1D are coupled. The schematic is for the BATS implementation, but is the same for HOTS except for the input data in the "Observational Data" box. The schematic is an edited version of one presented in Kern et al. (2024), now indicating that the optimization interface affects the BFM17, POM1D, and vertical velocity components of the biophysical model. The dashed line is used to indicate that DAKOTA treats the model as a black box and is only aware of the interface script controlling the model.

## 3  Model Description

The biophysical model used in this study, BFM17+POM1D (Smith et al., 2021), tracks 17 BGC tracers in a depth-dependent column of water over time. The time-rate of change for a state variable, $A_j$, in the model is described by

$$\frac{\partial A_j}{\partial t} = \frac{\partial A_j}{\partial t}\bigg|_{\text{bio}} - \left[W + W_E + v^{(\text{set})}\right]\frac{\partial A_j}{\partial z} + \frac{\partial}{\partial z}\left(K_H \frac{\partial A_j}{\partial z}\right), \tag{4}$$

where the various components of this equation are described in more detail in the following sections. Broadly, the three terms on the right-hand side of the equation correspond to BGC dynamics, vertical advection, and turbulent mixing, respectively. In this study, we consider the effect of including parameters for physical processes when tuning the model.

### 3.1  Biogeochemistry

The first term on the right-hand side of Eq. (4) is the rate of change due to BGC processes, which is modeled here using BFM17. This model is a simplified implementation of the complete BFM (Vichi et al., 2007), which itself uses a chemical functional family (CFF) approach to model BGC communities. This framework models different BGC communities by including or excluding ecosystem components based on their prevalence in the target community. Previous implementations of the model have





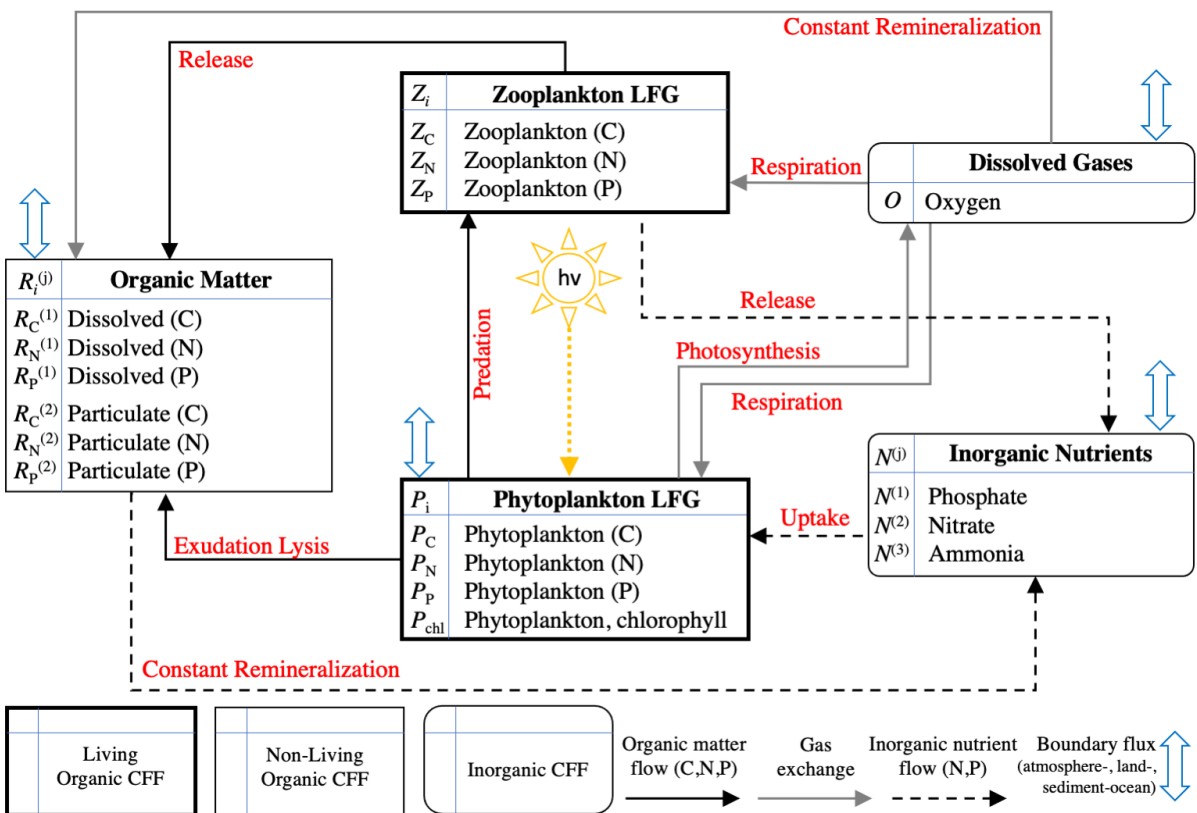

**Figure 2.** Schematic of the Biogeochemical Flux Model with 17 state variables (BFM17). The schematic lists all 17 state variables and shows the flux pathways between different chemical functional families (CFF). The schematic was originally presented in Smith et al. (2021).

included 56 state variables and the benthic system for studying the vertical structure of BGC communities in coastal regions. In
Smith et al. (2021), the 17 state variable implementation was developed to represent open-ocean oligotrophic BGC communities while being computationally affordable for implementation in high-resolution simulations of small to submesoscale ocean dynamics. Previous studies used BFM coupled with POM1D (Vichi et al., 2003; Mussap et al., 2016; Mussap and Zavatarelli, 2017a, b), but with the smaller implementation of BFM, we can also couple the BGC model to more complex physical models.

Compared to the full BFM, the formulation of BFM17 removes components based on assumptions appropriate for open
ocean oligotrophic conditions (i.e., the benthic system, iron cycle, silicate cycle, and bacteria) and introduces generalized living functional groups (LFGs) representing community average behavior instead of specific types of phytoplankton or zooplankton. Figure 2 shows all 17 state variables with parameterized flux pathways. The model includes a general phytoplankton LFG, a general zooplankton LFG, dissolved non-living organic matter, particulate non-living organic matter, three inorganic nutrients, and the dissolved gas oxygen.



Modeled BGC tracers include the constituent elements carbon, nitrogen, and phosphorus, with zooplankton, dissolved non-living organic matter, and particulate non-living organic matter each having three state variables. Phytoplankton has the additional state variable chlorophyll, due to the importance of chlorophyll dynamics and the availability of observational data for it. There are three inorganic nutrients included in BFM17: phosphate, nitrate, and ammonia. Each has one tracer element (phosphorus, nitrogen, and nitrogen, respectively), and therefore each contributes only one additional state variable to the model. The dissolved gas oxygen is tracked in the model, while carbon dioxide is treated as an infinite source or sink as needed.

A general discussion of BFM and the CFF approach is available in Vichi et al. (2007) and a full description of BFM17 and the included dynamics is available in Smith et al. (2021). With the formulation of BFM17 using generalized phytoplankton and zooplankton LFGs, re-evaluating parameter values is essential. The values that could be applied consistently for particular species must be tuned to a particular community to represent its average behavior. Although the parameters of the BGC model are estimated in Kern et al. (2024), in this study we begin with the baseline parameters of Smith et al. (2021). The parameter estimations here prioritize only the most sensitive BGC parameters instead of simultaneously estimating them all, and we additionally consider the effect of different approaches to including the physical model parameters in the estimation.

## 3.2   Vertical advection

The second term on the right-hand side of Eq. (4) describes the effect of advection on the vertical transport of BGC state variables. The parameterization in our biophysical model imposes a general circulation velocity $W$, and a vertical eddy velocity $W_E$. The final component is the state variable-dependent settling velocity, $v^{(\mathrm{set})}$, which is zero for all BGC variables except non-living particulate organic matter.

The vertical advection of BGC tracers is accomplished by two dynamical processes: seasonal general circulation and mesoscale eddies. The general circulation is assumed to result from Ekman pumping in a parameterization similar to the method employed by Bianchi et al. (2006). A vertical velocity profile is imposed that is zero at the surface and reaches a maximum value near the bottom of the Ekman layer, which is assumed here to correspond to the bottom of the model domain. The maximum velocity is calculated using Eq. (22) from Smith et al. (2021) as

$$W_{\mathrm{max}} = \hat{k} \cdot \boldsymbol{\nabla} \times \left( \frac{\boldsymbol{\tau}_w}{\rho f} \right), \tag{5}$$

where $\boldsymbol{\tau}_w$ is the wind stress, $\rho$ is the density, and $f$ is the Coriolis parameter. The curl of the wind stress is calculated using observational data. The resulting velocities are all negative, resulting in purely downward advection.

Mesoscale eddies provide vertical velocity perturbations on the time scale of weeks. These advection events are parameterized by multiplying the monthly profile of the vertical velocities of the general circulation by a randomly generated coefficient such that the maximum vertical eddy velocity is between 0.0 and 0.1 $\mathrm{md}^{-1}$. This is expressed by the equation

$$W_{E,m}^{(k)} = C_m^{(k)} W_m, \tag{6}$$

where $W_{E,m}^{(k)}$ is a set of 12 vertical velocity profiles. The general circulation is input monthly, while the eddy velocity profiles are input semi-monthly. The entire general circulation profile for a given month, $m$, is multiplied by a single coefficient, $C_m^{(k)}$.



The eddy velocity is assumed to have a 15 d timescale, so $k = 1$ corresponds to the set of vertical velocities used in the first half of each month and $k = 2$ for the second half. The vertical profile of the general circulation velocity at each time step is linearly interpolated from the monthly average profiles, while the eddy velocity profiles remain constant throughout their 15 d
period.

Although the 24 coefficients used to generate the eddy velocity profiles were initially generated randomly in Smith et al. (2021), here we treat them as parameters to be estimated. The initial coefficient values are included in Table 1, as are the upper and lower bounds used as constraints during the parameter estimation. Note that the initial values have been rounded and the upper and lower bounds were set to allow for maximum eddy velocities between 0.0 and 0.2 $\mathrm{md}^{-1}$, thereby relaxing the
original assumption limiting them to 0.1 $\mathrm{md}^{-1}$.

### 3.3 Turbulent diffusion

The final term on the right-hand side of Eq. (4) represents turbulent diffusion in the vertical direction. This part of the model has been adapted from the three-dimensional version of POM. Turbulent diffusion is parameterized by introducing a turbulent eddy viscosity, $K_M$, and a turbulent eddy diffusivity, $K_H$. The latter quantifies the effect of turbulent mixing on state variables
in the state variable transport equation.

The POM1D closure model is proposed in Mellor (2001) as a particular implementation of the one originally developed in Mellor and Yamada (1982). A discussion of the model as it is coupled to BFM17 is included in Smith et al. (2021). The closure model parameterizes the turbulent eddy viscosity and diffusivity as

$$K_M = q\ell S_M \,, \tag{7}$$

$$K_H = q\ell S_H \,, \tag{8}$$

where $q$ is the turbulent velocity, $\ell$ is the master length scale, and $S_M$ and $S_H$ are stability functions written as

$$S_M \left[1 - 9A_1 A_2 G_H\right] - S_H \left[\left(18A_1^2 + 9A_1 A_2\right) G_H\right] = A_1 \left[1 - 3C_1 - 6A_1/B_1\right] \,, \tag{9}$$

$$S_H \left[1 - (3A_2 B_2 + 18A_1 A_2) G_H\right] = A_2 \left[1 - 6A_1/B_1\right] \,. \tag{10}$$

The coefficients in the stability functions (i.e., $A_1, B_1, A_2, B_2, C_1$) are constants and

$$G_H = \frac{\ell^2}{q^2} \frac{g}{\rho_0} \frac{\partial \rho}{\partial z} \,. \tag{11}$$

Our implementation of BFM17+POM1D uses imposed temperature, $T$, and salinity, S, profiles from observational data to calculate the vertical density profile. Temperature and salinity are interpolated from monthly profiles at each time step. Given the vertical structure of the ocean state, the model calculates the vertical diffusivity using transport equations for the turbulence parameters $q$ and $\ell$. The governing equation for the turbulent kinetic energy, $q^2/2$, is

$$\frac{\partial}{\partial t}\left(\frac{q^2}{2}\right) = \frac{\partial}{\partial z}\left[K_q \frac{\partial}{\partial t}\left(\frac{q^2}{2}\right)\right] + K_M \left[\left(\frac{\partial U}{\partial z}\right)^2 + \left(\frac{\partial V}{\partial z}\right)^2\right] + \frac{g}{\rho_0} K_H \frac{\partial \rho}{\partial z} - \frac{q^3}{B_1 \ell} \,, \tag{12}$$



**Table 1.** List of randomly generated coefficient values for the baseline parameterization of semi-monthly vertical eddy velocity profiles from Smith et al. (2021) and Kern et al. (2024) for BATS and HOTS implementations, respectively, as well as the ranges considered in the present parameter estimation.

| Label | Parameter | Units | Description | BATS | | HOTS | |
|---|---|---|---|---|---|---|---|
| | | | | Baseline | Range | Baseline | Range |
| **Velocity perturbation coefficients $W_E^{(1)}$ (first half of month)** | | | | | | | |
| C.1 | $C_1^{(1)}$ | - | January | -4.12 | $-10.0 - 0.0$ | -0.99 | $-15.0 - 0.0$ |
| C.2 | $C_2^{(1)}$ | - | February | -2.28 | $-10.0 - 0.0$ | -0.46 | $-20.0 - 0.0$ |
| C.3 | $C_3^{(1)}$ | - | March | -6.64 | $-15.0 - 0.0$ | -1.33 | $-25.0 - 0.0$ |
| C.4 | $C_4^{(1)}$ | - | April | -4.06 | $-20.0 - 0.0$ | -0.81 | $-20.0 - 0.0$ |
| C.5 | $C_5^{(1)}$ | - | May | -13.77 | $-30.0 - 0.0$ | -2.75 | $-15.0 - 0.0$ |
| C.6 | $C_6^{(1)}$ | - | June | -6.94 | $-35.0 - 0.0$ | -1.39 | $-15.0 - 0.0$ |
| C.7 | $C_7^{(1)}$ | - | July | -16.37 | $-50.0 - 0.0$ | -3.27 | $-20.0 - 0.0$ |
| C.8 | $C_8^{(1)}$ | - | August | -73.15 | $-150.0 - 0.0$ | -14.63 | $-30.0 - 0.0$ |
| C.9 | $C_9^{(1)}$ | - | September | -636.91 | $-2895.0 - 0.0$ | -1.27 | $-40.0 - 0.0$ |
| C.10 | $C_{10}^{(1)}$ | - | October | -13.25 | $-65.0 - 0.0$ | -2.65 | $-30.0 - 0.0$ |
| C.11 | $C_{11}^{(1)}$ | - | November | -12.27 | $-20.0 - 0.0$ | -2.46 | $-20.0 - 0.0$ |
| C.12 | $C_{12}^{(1)}$ | - | December | -4.98 | $-10.0 - 0.0$ | -0.99 | $-15.0 - 0.0$ |
| **Velocity perturbation coefficients $W_E^{(2)}$ (second half of month)** | | | | | | | |
| C.13 | $C_1^{(2)}$ | - | January | -1.87 | $-10.0 - 0.0$ | -0.37 | $-15.0 - 0.0$ |
| C.14 | $C_2^{(2)}$ | - | February | -3.93 | $-10.0 - 0.0$ | -0.79 | $-20.0 - 0.0$ |
| C.15 | $C_3^{(2)}$ | - | March | -3.00 | $-15.0 - 0.0$ | -0.60 | $-25.0 - 0.0$ |
| C.16 | $C_4^{(2)}$ | - | April | -8.74 | $-20.0 - 0.0$ | -1.75 | $-20.0 - 0.0$ |
| C.17 | $C_5^{(2)}$ | - | May | -16.13 | $-30.0 - 0.0$ | -3.22 | $-15.0 - 0.0$ |
| C.18 | $C_6^{(2)}$ | - | June | 0.0 | $-35.0 - 0.0$ | 0.0 | $-15.0 - 0.0$ |
| C.19 | $C_7^{(2)}$ | - | July | -24.20 | $-50.0 - 0.0$ | -4.84 | $-20.0 - 0.0$ |
| C.20 | $C_8^{(2)}$ | - | August | -86.99 | $-150.0 - 0.0$ | -17.40 | $-30.0 - 0.0$ |
| C.21 | $C_9^{(2)}$ | - | September | -883.77 | $-2895.0 - 0.0$ | -1.77 | $-40.0 - 0.0$ |
| C.22 | $C_{10}^{(2)}$ | - | October | -38.06 | $-65.0 - 0.0$ | -7.61 | $-30.0 - 0.0$ |
| C.23 | $C_{11}^{(2)}$ | - | November | -5.99 | $-20.0 - 0.0$ | -1.20 | $-20.0 - 0.0$ |
| C.24 | $C_{12}^{(2)}$ | - | December | -0.71 | $-10.0 - 0.0$ | -0.14 | $-15.0 - 0.0$ |

while the governing equation for the master length scale, $\ell$, is

$$\frac{\partial}{\partial t}\left(q^2\ell\right) = \frac{\partial}{\partial z}\left[K_q\frac{\partial}{\partial t}\left(q^2\ell\right)\right] + E_1\ell K_M\left[\left(\frac{\partial U}{\partial z}\right)^2 + \left(\frac{\partial V}{\partial z}\right)^2\right] + E_1\ell\frac{g}{\rho_0}K_H\frac{\partial\rho}{\partial z} + \frac{q^3}{B_1}\tilde{W}. \tag{13}$$

In these equations, $K_q = \kappa K_H$ is a vertical diffusivity for the turbulent parameters and $\tilde{W} = \left[1 + E_2 l^2/\kappa^2(|z|^{-1} + |z - H|^{-1})^2\right]$ is a surface proximity function. Equation (13) and the surface proximity function include the nondimensional parameters $E_1$ and $E_2$, whose values have been included in Table A1.





**Table 2.** List of physical model parameters for the baseline implementation adopted from Smith et al. (2021).

| Label | Parameter | Units | Description | Baseline | Range |
|-------|-----------|-------|-------------|----------|-------|
| | | | **Velocity parameter** | | |
| P.1 | $v^{(\mathrm{set})}$ | $\mathrm{m\,d}^{-1}$ | Settling velocity of particulate detritus | 1.0 | $0.5 - 1.5$ |
| | | | **Boundary condition parameters** | | |
| P.2 | $\lambda_O$ | $\mathrm{m\,d}^{-1}$ | Relaxation constant for oxygen at bottom | 0.06 | $0.0 - 0.5$ |
| P.3 | $\lambda_{N^{(1)}}$ | $\mathrm{m\,d}^{-1}$ | Relaxation constant for phosphate at bottom | 0.06 | $0.0 - 0.5$ |
| P.4 | $\lambda_{N^{(2)}}$ | $\mathrm{m\,d}^{-1}$ | Relaxation constant for nitrate at bottom | 0.06 | $0.0 - 0.5$ |
| P.5 | $\kappa_{N^{(3)}}$ | $\mathrm{m^2 s}^{-1}$ | Relaxation diffusivity for ammonium at bottom | 0.05 | $0.0 - 0.5$ |
| | | | **POM1D parameters** | | |
| P.6 | $B1$ | - | Stability function coefficient | 16.6 | $10.0 - 20.0$ |

In the above equations, $g = 9.81\,\mathrm{m\,s}^{-1}$ is gravity, $\rho_0 = 1025\,\mathrm{kg\,m}^{-3}$ is the reference density, and $\kappa = 0.4$ is the Von Kármán constant. The horizontal velocity components, $U$ and $V$, are obtained by solving the momentum equations

$$\frac{\partial U}{\partial t} - fV = \frac{\partial}{\partial z}\left(K_M \frac{\partial U}{\partial z}\right), \tag{14}$$

$$\frac{\partial V}{\partial t} - fU = \frac{\partial}{\partial z}\left(K_M \frac{\partial V}{\partial z}\right), \tag{15}$$

where $f$ is the Coriolis parameter.

There are seven prescribed parameters in this model: the five coefficients included in the stability functions and the two non-dimensional parameters $E_1$ and $E_2$. The values of these parameters were determined in Mellor and Yamada (1982) and are included in Table A1. Since these coefficients parameterize the local turbulent dynamics, they are treated in this paper as unknown parameters that can be estimated. Ultimately, only one of these parameters (B1) is included in our parameter

estimation because of its relative importance in affecting model outputs.

### 3.4 Boundary conditions

The BGC model includes boundary conditions for the top and bottom of the water column. For all the state variables except oxygen, the flux at the surface is assumed to be zero. The flux at the bottom of the domain is also assumed to be zero for phytoplankton, zooplankton, dissolved non-living organic matter, and particulate non-living organic matter. The surface boundary

condition for oxygen is a parameterized air-sea gas flux ($\Phi_O$), defined as

$$K_H \frac{\partial O}{\partial z}\bigg|_{z=0} = \Phi_O, \tag{16}$$

which is based on the method developed in Wanninkhof (1992, 2014).



The bottom boundary conditions for oxygen, phosphate, and nitrate are described according to the equation

$$K_H \frac{\partial A_j}{\partial z}\bigg|_{z=z_{\text{end}}} = \lambda_j \left( A_j|_{z=z_{\text{end}}} - A_j^* \right), \tag{17}$$

where $\lambda_j$ is the corresponding relaxation velocity. The reference state variable $A_j^*$ is the value of the climatological field observed at the bottom boundary. The bottom boundary condition for ammonium is

$$K_H \frac{\partial A_j}{\partial z}\bigg|_{z=z_{\text{end}}} = \kappa_{N^{(3)}} \frac{\partial Q_N}{\partial z}\bigg|_{z=z_{\text{end}}}, \tag{18}$$

where $Q_N$ is referred to as the particulate organic nitrogen (PON), defined as the total particulate organic matter from phytoplankton, zooplankton, and the particulate non-living organic matter in terms of nitrogen; that is $Q_N = P_N + Z_N + R_N^{(2)}$.

The ammonia boundary condition returns nitrogen from depth in the form of ammonium, where particulate organic matter is recycled to the inorganic nutrient. The boundary condition is modulated by the diffusivity of ammonia, $\kappa_{N^{(3)}}$.

Although the top boundary condition for oxygen is considered to be well-defined, there are four additional uncertain parameters introduced by selecting these boundary conditions: the three relaxation velocities for oxygen, phosphate, and nitrate and the diffusivity for ammonia at the bottom boundary. The baseline values for these parameters are included in Table 2.

These four parameters were included in the optimization studies performed here to improve our representation of the physical processes parameterized by our model.

## 4  Physical Scenarios

In this study, we calibrate BFM17+POM1D for two physical scenarios that correspond to the mid-Atlantic and the subtropical Pacific. The former uses data from the Bermuda Testbed Mooring (Dickey et al., 2001) and BATS (Steinberg et al., 2001),

while the latter uses data from HOTS (Karl and Lukas, 1996). Both BATS and HOTS have long observational records for BGC and physical quantities. We averaged the observational data over all available years to obtain monthly climatological depth-dependent profiles that were then fitted to a 1 m grid for the top 150 m of the ocean. Measurements of salinity and temperature from BATS and HOTS, together with wind forcing data from the Scatterometer Climatology of Ocean Winds database (Risien and Chelton, 2008), were used to force BFM17+POM1D for the two scenarios. In both cases, BGC data for phytoplankton

chlorophyll, oxygen, nitrate, phosphate, and PON were used as target data for the biophysical model outputs. These five fields were selected because they are included in both the BATS and HOTS datasets, thereby providing a consistent set of inputs and outputs for the two scenarios. These physical scenarios were examined and described in detail in Kern et al. (2024), and in the following we provide a brief overview of each case.

### 4.1  Bermuda Atlantic time-series (BATS)

Observational and model data for our target fields at BATS are included in Fig. 3, showing phytoplankton chlorophyll, oxygen, nitrate, phosphate, and PON as functions of month and depth. Figure 3(a) is the observational data and Fig. 3(b) provides





**Figure 3.** BATS observational and model results for target-field concentration data for (from left-to-right) phytoplankton chlorophyll ($P_{\text{chl}}$), oxygen ($O$), nitrate ($N^{(2)}$), phosphate ($N^{(1)}$), and particulate organic nitrogen (PON). Row (a) is the reference observational data and subsequent rows correspond to model runs with the parameter values from Smith et al. (2021) (b), estimated BGC parameters only (c), estimated physical parameters only (d), sequentially estimated parameters (e), and simultaneously estimated parameters (f).





baseline model results from the original formulation of BFM17+POM1D in Smith et al. (2021). Figures 3 (c-f) are for the optimized parameter sets discussed in Section 7. Here, we compare the observational data to the baseline model configuration.

In the observational data shown in Fig. 3(a), the subsurface maximum in phytoplankton chlorophyll is observed between
255 50 m and 100 m for most of the year. A spring bloom is observed in the annual cycle of phytoplankton, resulting in increased growth as deeper mixing provides nutrients to the surface layer. During the summer months, the water column is more stratified, the subsurface maximum is pushed deeper, and the overall extent and magnitude of phytoplankton chlorophyll decreases. Oxygen generally fills the domain with lower concentrations at the top and bottom of the water column during the latter half of the year. Nitrate and phosphate have high concentrations at a depth of 150 m and lower concentrations in the upper water
column due to the biological pump. Both have a slight increase in concentration from November through January. The PON is largely confined to the top of the domain where the BGC community is most active. Similarly to the trend in phytoplankton, the subsurface maximum in PON shoals April through November.

Although the baseline model fails to predict the magnitudes of the observational tracer data, as shown in Fig. 3(b), it produces a similar behavior in the annual cycle for each of the target fields. The modeled phytoplankton chlorophyll concentrations
are higher than the observational values near the subsurface maximum throughout the year. However, following the annual trend, the model has increased vertical transport at the beginning of the year, with phytoplankton confined to the subsurface maximum through the summer and fall. Of the fields studied here, the modeled oxygen concentrations differ the most from the observational data, with a significantly under-predicted magnitude of the overall concentration. There is also a decrease in the oxygen concentration around 125 m for March through June, which is not present in the observational data. In agreement
with the observational values, the modeled concentrations of nitrate and phosphate are largely confined to the bottom of the domain. The concentration of nitrate is generally over-predicted, while phosphate is also generally over-predicted throughout the water column, except at the bottom of the domain from October through December. The modeled PON is confined to the upper portion of the domain as observed in the reference data, but the concentrations are over-predicted and do not reach as far down into the water column as the observational data.

**4.2 Hawaii Ocean time-series (HOTS)**

As with the BATS site, we show all observational and model data for the HOTS site in Fig. 4. The HOTS annual cycle has a similar pattern to that of BATS but with smaller seasonal changes, as shown in Fig. 4(a). Phytoplankton chlorophyll has a persistent subsurface maximum at 100 m depth, with higher concentrations from October through February. There are also increased concentrations below 100 m from March through July. There is a slight decrease in oxygen concentrations near the
280 surface from May through November, but the overall variation in concentration is relatively low. Nitrate and phosphate are confined to the bottom of the domain. They have increased vertical transport from November through February with additional upwelling events; the first is May through June for nitrate and April to May for phosphate, and the second is August to September for both. The PON is primarily in the upper portion of the domain. There is a slight increase in concentration from March through October between 25 m and 75 m.





**Figure 4.** HOTS observational and model results for target-field concentration data for (from left-to-right) phytoplankton chlorophyll ($P_{\text{chl}}$), oxygen ($O$), nitrate ($N^{(2)}$), phosphate ($N^{(1)}$), and particulate organic nitrogen (PON). Row (a) is the reference observational data and subsequent rows correspond to model runs with the parameter values from Smith et al. (2021) (b), estimated BGC parameters only (c), estimated physical parameters only (d), sequentially estimated parameters (e), and simultaneously estimated parameters (f).



Compared to the BATS site in Fig. 3, the results of the baseline model shown in Fig. 4(b) for HOTS diverge significantly from the observational data. However, it is important to note that the baseline parameter values used by Smith et al. (2021) were initially taken from a 56-state variable implementation of BFM for coastal locations. Some of the BGC and boundary control parameters in the baseline model were also manually tuned by Smith et al. (2021) specifically for the BATS site. As a result, it is not surprising that the observational and baseline model results do not match well for the HOTS location, since the baseline BFM17 parameters were not adjusted for this BGC community.

The baseline results in Fig. 4(b) show that the modeled phytoplankton, nitrate, phosphate, and PON have annual cycles similar to the observations, but the concentration magnitudes and community structure are poorly predicted. Phytoplankton chlorophyll is over-predicted throughout the year and the subsurface maximum is not as deep in the water column as the observational data, being closer to 75 m than the observed 100 m. Nitrate is severely over-predicted, while phosphate is under-predicted to a lesser degree. PON is severely over-predicted and does not extend as deep into the water column as in the observational data. The most significant differences are between the observational and simulated oxygen fields. Neither the annual cycle nor the concentrations are close to those in the observational data.

By comparing the observational and baseline model results for BATS and HOTS, we observe similarities in the trends in the observed and predicted annual cycles of the target fields. This suggests that BFM17+POM1D is capable of successfully representing the target BGC communities. However, there are significant differences, which implies that the model requires tuning to successfully represent a biogeochemical community. In Kern et al. (2024), the complete set of BGC parameters in BFM17 was optimized to improve the model results, both individually and simultaneously at BATS and HOTS. Considerable improvements in the model agreement were obtained for HOTS in particular. In the present study, we examine whether we can increase the accuracy of the model by simultaneously estimating key physical and BGC parameters. Simultaneous physical and BGC parameter estimation should produce more realistic BGC and physical parameter estimates than sequential physical-BGC parameter estimation, which can produce BGC parameter estimates that compensate for biases in physical parameter estimates, and vice versa.

## 5 Parameter Sensitivity Analysis

Kern et al. (2024) primarily focused on estimating parameters in the BGC model. Here, we also estimate 30 parameters from the physical model, a majority of which were not previously considered. The parameters included from the physical model are 1 turbulent mixing parameter, 1 sinking velocity, 4 boundary control parameters, and 24 coefficients for the vertical eddy velocity profiles. The sinking velocity and boundary control parameters were included by Kern et al. (2024), but here they are included specifically as part of the physical model. Since there are already 30 identified physical model parameters to be estimated, we opted to reduce the full set of 46 BGC parameters in BFM17 to manage the computational requirements of the estimation problem.

The greater the number of parameters, the more complex the objective function space. This is an important consideration, as the parameter estimation methodology implemented here employs gradient-based optimization. For gradient-based optimiza-





**Table 3.** List of BFM17 parameters controlling the marine ecosystem dynamics in the model selected to be included in the parameter estimation study.

| No. | Parameter | Units | Description | Baseline value | Range |
|------|-----------|-------|-------------|----------------|-------|
| P.7 | $\varepsilon_{\mathrm{PAR}}$ | - | Fraction of photosynthetically available radiation | 0.4 | 0.25-0.75 |
| P.8 | $\lambda_w$ | $\mathrm{m}^{-1}$ | Background attenuation coefficient | 0.0435 | 0.03-0.05 |
| P.9 | $c_P$ | $\mathrm{m}^{-2}\,(\mathrm{mg\,Chl})^{-1}$ | Chlorophyll-specific light absorption coefficient | 0.03 | 0.005-0.045 |
| P.10 | $b_P$ | $\mathrm{d}^{-1}$ | Basal specific nutrient-stress lysis rate | 0.05 | 0.005-0.075 |
| P.11 | $\phi_{\mathrm{N}}^{(\mathrm{opt})}$ | $\mathrm{mmol\,N}\,(\mathrm{mg\,C})^{-1}$ | Optimal nitrogen quota | $1.26\times10^{-2}$ | $1.0\times10^{-4}$-$5.0\times10^{-2}$ |
| P.12 | $\phi_{\mathrm{P}}^{(\mathrm{opt})}$ | $\mathrm{mmol\,P}\,(\mathrm{mg\,C})^{-1}$ | Optimal phosphorous quota | $7.86\times10^{-4}$ | $1.0\times10^{-4}$-$1.0\times10^{-3}$ |
| P.13 | $\alpha_{\mathrm{chl}}^{(0)}$ | $\mathrm{mg\,C}\,(\mathrm{mg\,Chl})^{-1}\mu\,\mathrm{E}^{-1}\mathrm{m}^2$ | Maximum light utilization coefficient | $1.52\times10^{-5}$ | $5.0\times10^{-6}$-$5.0\times10^{-5}$ |
| P.14 | $\theta_{\mathrm{chl}}^{(0)}$ | $\mathrm{mg\,Chl}\,(\mathrm{mg\,C})^{-1}$ | Maximum chlorophyll-to-carbon quota | 0.016 | 0.005-0.05 |
| P.15 | $r_Z^{(0)}$ | $\mathrm{d}^{-1}$ | Potential specific growth rate | 2.0 | 1.0-7.5 |
| P.16 | $d_Z$ | $\mathrm{d}^{-1}$ | Specific mortality rate | 0.05 | 0.025-0.1 |
| P.17 | $\eta_Z$ | - | Assimilation efficiency | 0.5 | 0.05-0.55 |
| P.18 | $h_Z^{(F)}$ | $\mathrm{mg\,C\,m}^{-3}$ | Michaelis constant for total food ingestion | 200.0 | 50.0-500.0 |

tion, we have to estimate the gradient in each dimension in the objective function space. Each additional parameter introduces a dimension along which we have to approximate the gradient with finite differencing, requiring two more model evaluations.
Also, the more complex the objective space, the easier it is to fail to converge to a solution or to converge to a poorly performing local minimum. We therefore reduce the set of BGC parameters; the resulting parameter set is still sufficient to allow a comparison of sequential versus simultaneous optimization of BGC and physical models, which is the primary focus of the present study.

To reduce the set of parameters to be estimated in the BGC model, we performed a one-at-a-time sensitivity study to
325 determine the relative sensitivity of BFM17 to each parameter. Each parameter is independently perturbed by 5% from its nominal value. The model is tested with a positive and negative perturbation, except where the perturbed value exceeds the upper or lower bound of the parameter range. The nominal parameter values and parameter limits are provided in Table 3 for those ultimately selected for the estimation study and in Table A1 for those excluded. The objective function, defined in Eq. (2), is calculated for each perturbed case, with the maximum of the positively and negatively perturbed cases taken as the
330 sensitivity metric, $S_p$, for each parameter. The parameters are compared using a relative importance, $\hat{S}_p$, defined as

$$\hat{S}_p = S_p/S_p^{(\mathrm{max})}, \tag{19}$$

where $S_p^{(\mathrm{max})}$ is the maximum sensitivity factor across parameters for a given physical scenario.

The results of the sensitivity analysis are shown in Fig. 5 for the BATS and HOTS locations. Relative sensitivities are shown for each parameter, with the parameters ordered according to relative importance values (from highest to lowest). Figure 5
compares the contribution of each target field to the final relative importance metric, as indicated by the portion of the total normalized differences that come from each field. The most important parameters are those with the highest relative importance values. We separated the parameters into two groups using a threshold of $\hat{S}_p = 0.1$, which is indicated in Fig. 5 by dashed lines,





**Figure 5.** Results of a one-at-a-time sensitivity study performed for the BATS (a) and HOTS (b) locations. The parameters are ranked based on their relative importance for each location, with the values shown on the bar plot. A dotted line is included on both plots to indicate where the relative importance values fall below 0.1.

with the parameters on the left having relative importance greater than the threshold and the parameters on the right less than the threshold. The 0.1 threshold is used in this study because we would like no more than 20 additional BGC parameters to keep the total number of parameters estimated similar to the number included in Kern et al. (2024). Furthermore, the twin simulation studies presented by Kern et al. (2024) suggest that we should be able to recover parameters with relative importance values greater than 0.01 for BFM17+POM1D. Our threshold of 0.1 is therefore conservative. Ultimately, fifteen parameters met the threshold condition for BATS, while 16 parameters met the condition for HOTS. We chose to use the minimum number of common parameters between the two locations (i.e., the intersection set) since we would like to optimize a consistent set of parameters for both locations and the intersection set includes most of the important parameters at each location.

In total, there are 13 parameters in the intersection set, of which 12 will be used in our estimation studies. The selected parameters are presented in Table 3. The zooplankton feeding threshold, $\mu_Z$, also met the criteria to be included in the estimation




study. However, since the parameterization of zooplankton production (Smith et al., 2021, Eq. (A34)) uses the potential specific growth rate ($r_Z^{(0)}$), the Michaelis constant for total food ingestion ($h_Z^{(F)}$), and the feeding threshold ($\mu_Z$), only the two most
350 important of the three parameters (as expressed by the relative importance) are retained for subsequent calibration studies.

## 6 Twin Simulation Experiment

Before proceeding with the parameter estimation studies, the methodology and its implementation were tested using a Twin Simulation Experiment (TSE), as in Spitz et al. (1998), Athias et al. (2000), Kidston et al. (2011), Oliver et al. (2022), and Kern et al. (2024). In a TSE, we attempt to recover the values of known parameters from a perturbed initial state. The target
data are synthetic and generated using the model with the baseline parameter values. The TSE presented here was performed for the BATS location using the 12 selected BGC parameters from the previous section. Using the perturbed initial values, we tested whether the applied method was able to correctly predict the baseline parameter values. The TSE parameters and their target values (i.e., the baseline values) are included in Table 3.

The initial parameter values were set by increasing the baseline parameter values by 10%, consistent with the tests performed
by Kern et al. (2024). We then applied the parameter estimation methodology to recover the baseline values, giving the results shown in Fig. 6, where we calculate the difference between the normalized parameter value tested, $\hat{p}_i$, and the normalized

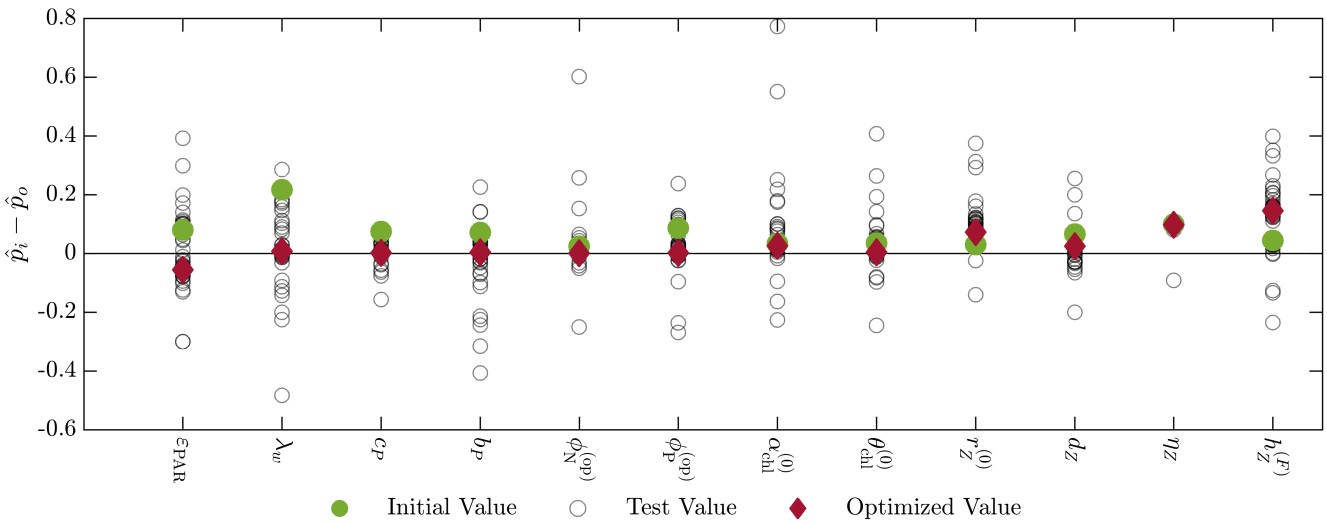

**Figure 6.** Results of the twin simulation experiment with the 12 selected BGC parameters included in the parameter estimation study. The initial values (green dot), final value (red diamond), and intermediate test values (empty circles) are shown as the difference between the normalized parameter value ($\hat{p}_i$) and the normalized nominal parameter value ($\hat{p}_o$). The normalization is between 0 and 1 based on the upper and lower bounds of the parameter value ranges presented in Table 3.





baseline parameter value, $\hat{p}_0$, for each parameter. The parameter values are normalized to a domain between 0 and 1 using the lower and upper limits included in the final column of Table 3.

Figure 6 shows that the estimated parameter values are all close to the baseline values. Of the 12 parameters tested, 9 have

365 moved towards the baseline value, while 1 does not change. As noted in the previous section, the remaining two parameters, $r_Z^{(0)}$ and $h_Z^{(F)}$, control the same zooplankton equation in the BGC model. Therefore, these two parameters are able to adjust to compensate for each other, so there is no major concern if they do not converge to the exact baseline values.

The results of the TSE indicate that the parameter estimation methodology can be successfully applied to the model to improve the agreement with the target data. The results of the TSE also confirm that the optimization toolbox and the model

have been correctly connected to each other. Most importantly, the results indicate that we have selected a reasonable set of parameters from the BGC model to proceed with the rest of the study.

## 7 Parameter Estimation for Observed Time Series

In the following, we describe the results of sequential and simultaneous estimations of the BGC and physical model parameters in BFM17+POM1D for the BATS and HOTS observed time series. The results of all parameter estimations performed are

375 shown along with the corresponding observational data in Figs. 3 and 4 for BATS and HOTS, respectively. We also show RMSD values for each model, relative to the observational data, in Table 4.

Sequential calibration studies were performed by applying the parameter estimation methodology first to the parameters of the physical model, beginning with the baseline parameters in Smith et al. (2021). The resulting state variable fields are shown in Figs. 3(d) and 4(d) for BATS and HOTS, respectively. The physical parameters that produce the best overall agreement are

380 then used as the starting point in another calibration of the 12 BGC parameters identified in Section 5. The resulting model fields are shown in Figs. 3(e) and 4(e). In each of the initial (physical) and secondary (BGC) estimation steps, sampling is performed for $N_{\mathrm{samp}} = 10,000$ randomly generated parameter sets. Then, the gradient-based optimization algorithm is applied to the $N_{\mathrm{top}} = 20$ best-performing parameter sets. For reference, we also show results from the estimation of only BGC model parameters in Figs. 3(c) and 4(c). We do not consider here the sequential case where the BGC parameters are estimated first,

since the baseline parameters from Smith et al. (2021) were already manually calibrated to give good agreement with the results from BATS.

The simultaneous calibration case estimates the complete set of parameters of the coupled biogeochemical and physical model at the same time, starting from the baseline parameters in Smith et al. (2021). The fields resulting from the simultaneous calibration studies are presented in Figs. 3(f) and 4(f) for BATS and HOTS, respectively. For these studies, initial sampling

is performed for $N_{\mathrm{samp}} = 20,000$ random parameter sets. The sampling in this case is intended to match the total number of samples applied to obtain the final solution in the sequential calibration case. We still apply the optimization algorithm to the $N_{\mathrm{top}} = 20$ best-performing parameter sets.

The number of randomly sampled sets of parameters for sequential and simultaneous optimization cases is determined primarily on the basis of available computational resources. The number of parameter sets used to initialize optimization runs



**Table 4.** Field specific and total normalized RMSD values for the baseline and calibrated models at the BATS and HOTS locations. The calibration results include the sequential and simultaneous calibration cases, plus BGC-only and physics-only cases for reference.

| Field | BATS | | | | | HOTS | | | | |
|---|---|---|---|---|---|---|---|---|---|---|
| | Baseline | Sequential | Simultaneous | BGC only | Physics only | Baseline | Sequential | Simultaneous | BGC only | Physics only |
| Chlorophyll | 1.84 | 0.84 | 0.73 | 1.01 | 1.90 | 2.81 | 0.38 | 0.83 | 6.21 | 2.97 |
| Oxygen | 5.49 | 4.10 | 1.82 | 5.00 | 3.61 | 27.20 | 7.81 | 1.42 | 39.96 | 3.95 |
| Nitrate | 0.75 | 0.90 | 0.87 | 0.51 | 1.03 | 130.47 | 0.81 | 0.90 | 75.38 | 0.66 |
| Phosphate | 0.82 | 0.64 | 0.40 | 0.84 | 0.79 | 1.43 | 0.57 | 0.59 | 1.80 | 1.31 |
| PON | 5.80 | 1.11 | 0.92 | 1.01 | 2.51 | 7.54 | 1.84 | 0.84 | 5.78 | 4.04 |
| Total | 14.70 | 7.59 | 4.74 | 8.37 | 9.84 | 169.45 | 11.41 | 4.58 | 129.13 | 12.93 |

is based on similar reasoning; however, the analysis included in Appendix B of Kern et al. (2024) suggests $N_{\text{top}} = 20$ should be sufficient.

Comparisons of simultaneous and sequential parameter estimations are performed separately for the BATS and HOTS sites. Each site is modeled using the appropriate BATS or HOTS forcing data for temperature, salinity, and winds, with site-specific initial BGC and nitrate boundary values, as well as vertical eddy parameterization coefficients, as indicated in Table 1. The number of sites for the objective function given in Eq. (2) is therefore $N_{\text{s}} = 1$. In the following parameter estimation studies, we use phytoplankton chlorophyll, oxygen, nitrate, phosphate, and PON as our target fields, so $N_{\text{v}} = 5$.

### 7.1 Bermuda Atlantic Time Series

Figure 3 and Table 4 show that the calibrated case that only includes the 12 most sensitive BGC parameters was unable to significantly improve agreement with the observational data for BATS, compared to the baseline model. Similarly, only estimating the physical parameters (as the first calibration in the sequential study) produced only moderate improvements. A comparison of fields across the calibration cases demonstrates that the simultaneous optimization of the physical and BGC parameters produces the best results, with each of the variable fields generally in better agreement with the observational data in the simultaneous estimation case. Nitrate is an exception, where the optimized results under-predict the observed concentrations at the bottom boundary. This trend is also evident in the normalized RMSD values calculated for each of the target fields.

The results of the BGC only and sequential calibration cases were similar in overall agreement, and sequential calibration outperformed the previous two calibration methods. The field-specific normalized RMSD values show that better agreement is obtained for phytoplankton chlorophyll, oxygen, and phosphate. Improvements in chlorophyll and oxygen can also be clearly observed in the field results.

The coefficients for the vertical eddy velocities are an important component of the parameter estimation. Figure 7 shows the complete vertical velocity profiles of the baseline parameterization of mesoscale eddies. Figure 7(a) shows the velocity profiles applied to the first 15 days of each month, and Fig. 7(b) shows the profiles for the second 15 days. Figure 7(c) shows the full time series using the maximum velocity of each profile, which is always located at the bottom of the water column. The estimated results are shown in Fig. 7(c) using a moving average to show the trend in the velocities throughout the year.

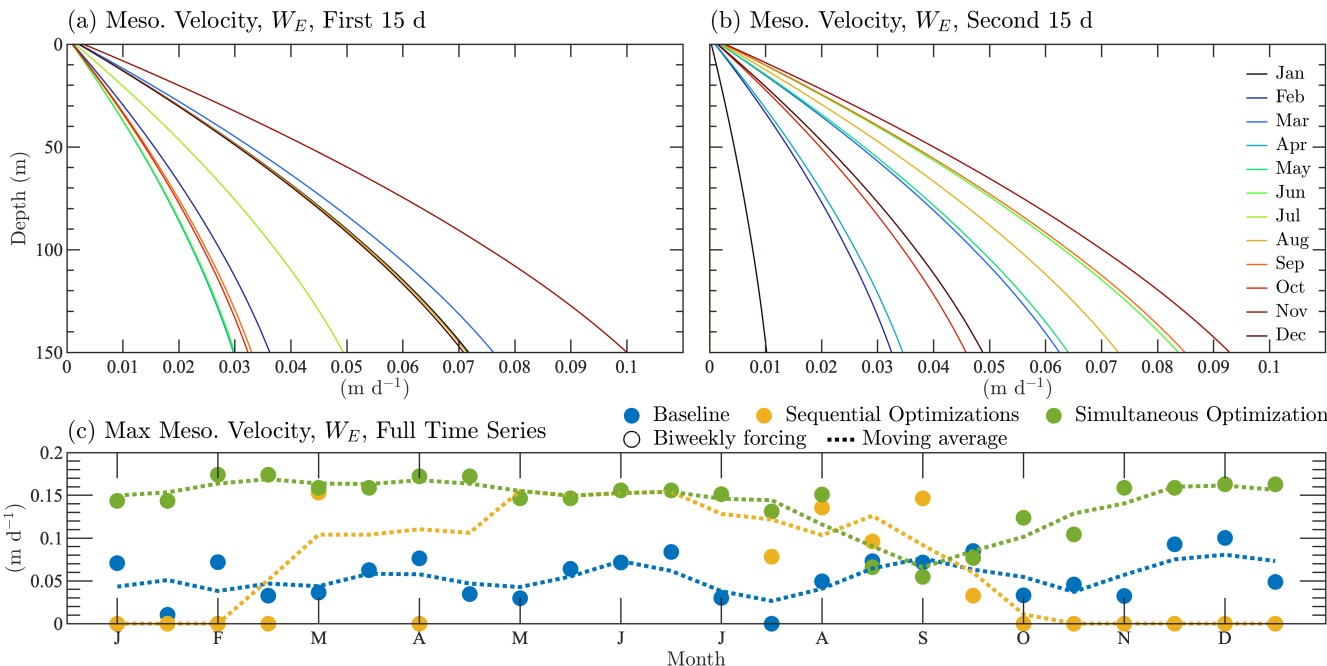

**Figure 7.** Parameterization of vertical velocity profiles from mesoscale eddies for the BATS implementation. Panels (a) and (b) are the full semi-monthly velocity profiles applied to the first and second halves of each month, respectively. The profiles correspond to the baseline case. In panel (c), the maximum velocity values are shown throughout the annual cycle, including data from the baseline case, the sequential calibration, and the simultaneous calibration.

Figure 7(c) shows that the baseline parameterization is randomly distributed between 0 and $0.1 \ \mathrm{md}^{-1}$ throughout the year. There is a slight shift towards higher velocities at the end of the year after August. As a result of relaxing the upper limit to $0.2 \ \mathrm{md}^{-1}$, both the sequentially and simultaneously calibrated velocity profiles produce higher velocities, but with different trends. The sequential calibration case increases vertical velocities from March through the end of August. From October through February, the velocities are estimated to be $0 \ \mathrm{md}^{-1}$.

The simultaneous estimation produces a more reasonable annual cycle in vertical velocities, contributing to better agreement with the observational values at BATS. Vertical mesoscale velocities are predicted to be higher for most of the year than in the baseline case, but with a clear decrease in vertical velocities from July through the end of October. These trends match expectations, since there is more vertical transport associated with mesoscale eddies (Mahadevan and Archer, 2000; Salmon et al., 2015) during the fall and winter when the eddies are more abundant (Aguedjou et al., 2019).

## 7.2  Hawaii Ocean Time Series

At the HOTS location, Fig. 4 and Table 4 show that the calibrated case only including the 12 most sensitive BGC parameters was unable to significantly improve agreement with the observational data. However, in this case, there are significant improvements




compared to the BATS case for the estimation of the physical parameters only, which is followed by moderately improved agreement from the subsequent calibration of the BGC model parameters.

Despite significant improvements in the sequential calibration case, the best overall agreement was achieved by simultane-
ously optimizing the BGC and the physical model parameters, as in the BATS scenario. All five variable fields show improved results, reproducing the annual cycle. The phytoplankton chlorophyll and nitrate are still over-predicted in some portions of the water column and the PON has a subsurface maximum near 100 m instead of the observed 25-50 m, but in all cases the normalized RMSD has decreased. With the exception of phytoplankton, all simultaneous calibration results are at least on par with those of the sequential calibration methodology. The simultaneous calibration case ultimately performs best by getting significantly better results in oxygen (with a normalized RMSD of 1.42 compared to 7.81, as shown in Table 4) and PON (0.84 compared to 1.84), resulting in a total normalized RMSD of 4.58 compared to 11.41.

Figure 8 shows the annual variability in mesoscale eddy vertical velocity profiles for the baseline, sequential calibration, and simultaneous calibration cases, following the format of Fig. 7. The baseline case set the vertical eddy velocities very low, with the majority less than $0.05 \ \mathrm{md}^{-1}$. The values are especially low from January to March, after which they start to increase, with the largest values occurring in August and September. The sequential calibration increases velocities throughout the year, while driving some values to $0 \ \mathrm{md}^{-1}$, especially from mid-January through the beginning of April. The variability predicted by the simultaneous calibration case is the most reasonable, with the annual cycle better matching expectations for the HOTS location. Vertical velocities throughout the year are higher, with a majority of values between $0.15 \ \mathrm{md}^{-1}$ and $0.2 \ \mathrm{md}^{-1}$. Overall, these results are reasonable and are similar to the trend observed at BATS, with some differences in the timing and degree of seasonal trends.

## 8   Conclusions

We have estimated parameter values for the coupled BGC and physical model BFM17+POM1D using a multi-step automatic methodology that was previously developed in Kern et al. (2024) to efficiently tune a large number of model parameters. Our objective in this study has been to find a methodology for estimating the parameters that will most accurately represent the observed data from target BGC fields. We found that it was necessary to include the estimation of the physical model parameters to get the best agreement with the target fields. The cases we have performed address questions about how to efficiently and effectively incorporate BGC and physical parameter sets into a single parameter estimation.

In our study, we examine for the first time the best approach to apply the parameter estimation methodology to both com-
ponents of a coupled BGC and physical model. We tested two approaches: the first was to sequentially calibrate the physical parameters and then the BGC parameters. It should be noted that this approach could also be applied in an iterative framework where, after estimating both sets of parameters, the entire sequential estimation procedure would be repeated. Here, we have performed the first iteration of such a process.

The second approach examined here involved the simultaneous estimation of both physical and BGC parameters. For the BATS scenario, the simultaneous estimation outperformed the sequential estimation for all target fields. The general disagree-



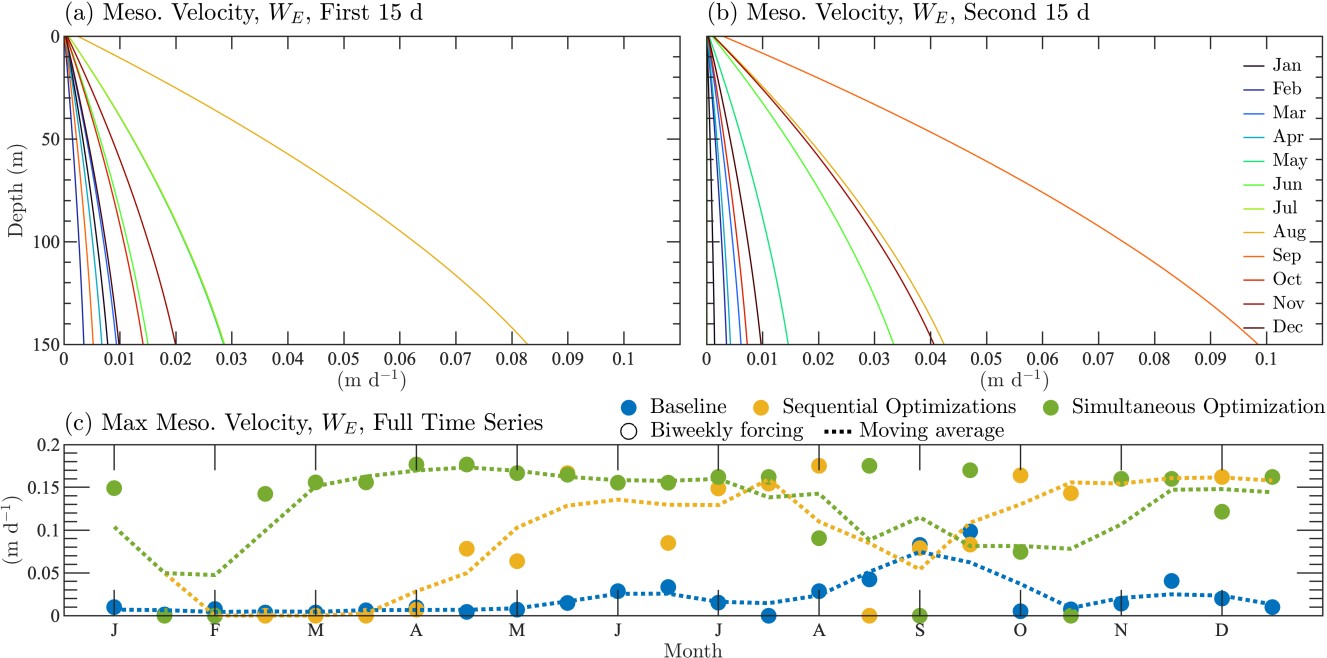

**Figure 8.** Parameterization for vertical velocity profiles from mesoscale eddies for the HOTS implementation. Panels (a) and (b) are the full semi-monthly velocity profiles applied to the first and second hafts of each month, respectively. The profiles correspond to the baseline case. In panel (c), the trend in vertical velocity profiles is shown using the maximum velocity throughout the annual cycle, including data from the baseline case, the sequential calibration, and the simultaneous calibration.

ment between the model results and observational data was 30% less in the simultaneous estimation than in the sequential estimation. For the HOTS scenario, the overall model performance was much better after the simultaneous estimation than in the sequential case. The overall disagreement from the sequential results was more than a factor of two higher than the simultaneous results.

An important outcome of the parameter estimation studies was more realistic trends in the vertical eddy velocities. The

simultaneous parameter estimation for BATS produced an annual cycle in vertical eddy velocities that was more realistic. Similarly, the simultaneous parameter estimation for HOTS produces a more reasonable trend. However, in the latter case, there was decreased activity later in the year.

It is important to clarify that the need to simultaneously calibrate both components of the biophysical model arises from the specific formulation of the physical model and parameterizations in POM1D. The model has imposed temperature and salinity

profiles from the observational time series, while vertical velocities are imposed from a parameterization assuming a general circulation profile based on wind stress values and an assumed Ekman depth. The mesoscale eddy velocities are semi-monthly perturbed versions of the monthly general circulation profiles. We only have observational BGC data to perform the estimation, which necessitates the simultaneous estimation of both model components. In other contexts, where there are passive tracers





that can be used to estimate the physical model separately from the BGC model, it could be practical and beneficial to calibrate
the physical model first and subsequently the BGC model. Being able to initially calibrate the physical model, separate from
the BGC model, would likely make the BGC parameter calibration converge faster.

Although this study specifically examines the best way to incorporate physical and BGC models into a single calibration,
it also serves as a demonstration of a meta-algorithm for performing calibrations of high-dimensional models with local opti-
mization approaches. In Kern et al. (2024), we addressed the problem by developing a methodology, summarized in Section 2,
capable of estimating the full set of BGC parameters. In this study, we include a similar number of parameters, but not all of
the BGC parameters. As a result, the meta-algorithm demonstrated can be summarized as:

1. Test model sensitivity to input parameters;

2. Select parameters for estimation;

3. Probe the reduced parameter space;

4. Perform local optimizations in best regions;

5. Select the best performing parameter set as the final solution.

The particular methodology applied in this paper is a specific implementation of this general framework. For model sensitivity,
we perform one-at-a-time parameter perturbations. We then select the parameters that are most sensitive for different observa-
tional time series. We probed the parameter spacing using randomly generated parameter sets with a Latin hybercube sampling
algorithm from DAKOTA. Finally, we use a quasi-Newton algorithm from the Opt++ library in DAKOTA to perform local
optimizations and select the best parameter set from their results.

Although the tools used here have been shown to work well for the present parameter estimations, all can be changed to fit the
problem being addressed, the needs of the user, and the available computational resources. For example, a more sophisticated
sensitivity analysis could be performed using an active subspace analysis. Instead of randomly sampling the parameter space,
the user might also take advantage of a more direct approach, such as a truncated evolutionary algorithm. Other optimization
algorithms like conjugate gradient or constrained optimization by linear approximation could be appropriate local optimization
algorithms for other applications.

*Code and data availability.* The codes and data necessary to reproduce the results in this paper have been archived on Zenodo. The data and
scripts required to reproduce the figures included in this paper are archived at Kern et al. (2025a). The code for performing the parameter
estimations is archived at Kern et al. (2025b). The updated implementation of the BFM17+POM1D code (originally developed by Smith
et al., 2021) used in the parameter estimations cases presented here is archived in Kern et al. (2025c).

**Appendix A**





**Table A1.** List of values for BFM17 parameters not included in the calibration.

| No. | Parameter | Value | Units | Description |
|---|---|---|---|---|
| | | *Phytoplankton Parameters* | | |
| 1 | $c_{R(2)}$ | $0.5 \times 10^{-4}$ | $\text{m}^2\,(\text{mg C})^{-1}$ | C-specific attenuation coefficient of particulate detritus |
| 2 | $r_P^{(0)}$ | 1.0 | $\text{d}^{-1}$ | Maximum specific photosynthetic rate |
| 3 | $d_P^{(0)}$ | 0.025 | $\text{d}^{-1}$ | Maximum specific nutirent-stess lysis rate |
| 4 | $h_P^{(\text{N,P})}$ | 0.05 | - | Nutrient-stress threshold |
| 5 | $\beta_P$ | 0.025 | - | Excreted fraction of primary production |
| 6 | $\gamma_P$ | 0.025 | - | Activity respiration fraction |
| 7 | $a_P^{(\text{N})}$ | 0.0375 | $\text{m}^3(\text{mg C})^{-1}\,\text{d}^{-1}$ | Specific affinity constant for nitrogen |
| 8 | $h_P^{(\text{N})}$ | 1.0 | $\text{mmol NH}_4\,\text{m}^{-3}$ | Half-saturation constant for ammonium uptake |
| 9 | $\phi_N^{(\text{min})}$ | $6.079 \times 10^{-3}$ | $\text{mmol N}\,(\text{mg C})^{-1}$ | Minium nitrogen quota |
| 10 | $\phi_N^{(\text{max})}$ | $1.49\phi_N^{(\text{opt})}$ | $\text{mmol N}\,(\text{mg C})^{-1}$ | Maximum nitrogen quota |
| 11 | $a_P^{(\text{P})}$ | $3.75 \times 10^{-3}$ | $\text{m}^3\,(\text{mg C})^{-1}\text{d}^{-1}$ | Specific affinity constant for phosphorous |
| 12 | $\phi_P^{(\text{min})}$ | $2.175 \times 10^{-4}$ | $\text{mmol P}\,(\text{mg C})^{-1}$ | Minimum phosphorous quota |
| 13 | $\phi_P^{(\text{max})}$ | $1.0\phi_P^{(\text{opt})}$ | $\text{mmol P}\,(\text{mg C})^{-1}$ | Maximum phosphorous quota |
| 14 | $l_P^{(\text{sink})}$ | 0.25 | - | Nutrient stress threshold for sinking |
| 15 | $w_P^{(\text{sink})}$ | 0.25 | $\text{m d}^{-1}$ | Maximum sinking velocity |
| | | *Zooplankton Parameters* | | |
| 16 | $b_Z$ | 0.01 | $\text{d}^{-1}$ | Basal specific respiration rate |
| 17 | $d_Z^{(0)}$ | 0.375 | $\text{d}^{-1}$ | Oxygen-dependent specific mortality rate |
| 18 | $\beta_Z$ | 0.375 | - | Fraction of activity excretion |
| 19 | $\varepsilon_Z^{(C)}$ | 0.5 | - | Partition between dissolved and particulate excretion of C |
| 20 | $\varepsilon_Z^{(N)}$ | 1.0 | - | Partition between dissolved and particulate excretion of N |
| 21 | $\varepsilon_Z^{(P)}$ | 1.0 | - | Partition between dissolved and particulate excretion of P |
| 22 | $h_Z^{(O)}$ | 0.75 | $\text{mmol O}_2\,\text{m}^{-3}$ | Half saturation for zooplankton processes |
| 23 | $\mu_Z$ | 40.0 | $\text{mg C}\,\text{m}^{-3}$ | Feeding Threshold |
| 24 | $\varphi_{\text{P}}^{(\text{opt})}$ | $7.0 \times 10^{-4}$ | $\text{mmol P}\,(\text{mg C})^{-1}$ | Optimal phosphorous quota |
| 25 | $\varphi_{\text{N}}^{(\text{opt})}$ | $1.0 \times 10^{-2}$ | $\text{mmol N}\,(\text{mg C})^{-1}$ | Optimal nitrogen quota |
| 26 | $\delta_{Z,P}$ | 1.0 | - | Availability of phytoplankton to zooplankton |
| | | *Non-living Organic Parameters* | | |
| 27 | $\Lambda_{N3}^{(\text{nit})}$ | $5.0 \times 10^{-3}$ | $\text{d}^{-1}$ | Specific nitrification rate at $10^\circ\text{C}$ |
| 28 | $h_N^{(O)}$ | 10.0 | $(\text{mmol O}_2)\,\text{m}^{-3}$ | Half saturation for chemical processes |
| 29 | $\xi_{\text{CO}_2}$ | 0.05 | $\text{d}^{-1}$ | Specific remineralization rate of particulate carbon |
| 30 | $\xi_{N(1)}$ | 0.05 | $\text{d}^{-1}$ | Specific remineralization rate of particulate phosphorous |
| 31 | $\xi_{N(3)}$ | 0.05 | $\text{d}^{-1}$ | Specific remineralization rate of particulate nitrogen |
| 32 | $\zeta_{\text{CO}_2}$ | 0.45 | $\text{d}^{-1}$ | Specific remineralization rate of dissolved carbon |
| 33 | $\zeta_{N(1)}$ | 0.22 | $\text{d}^{-1}$ | Specific remineralization rate of dissolved phosphorous |
| 34 | $\zeta_{N(3)}$ | 0.05 | $\text{d}^{-1}$ | Specific remineralization rate of dissolved nitrogen |
| | | *Additional POM Parameters* | | |
| 35 | $A_1$ | 0.92 | $\text{d}^{-1}$ | Stability function coefficient |
| 36 | $A_2$ | 0.74 | $\text{d}^{-1}$ | Stability function coefficient |
| 37 | $B_2$ | 10.1 | $\text{d}^{-1}$ | Stability function coefficient |
| 38 | $C_1$ | 0.08 | $\text{d}^{-1}$ | Stability function coefficient |
| 39 | $E_1$ | 1.8 | $\text{d}^{-1}$ | Turbulence parameter equation coefficient |
| 40 | $E_2$ | 1.33 | $\text{d}^{-1}$ | Turbulence parameter equation coefficient |





**Table A2.** List calibrated model parameters and coefficients for BATS and HOTS implementations respectively. If the bounds vary between cases which is only true for the velocity coefficients, the HOTS parameter is in parenthesis.

| No. | Parameter | Units | Range | BATS | HOTS |
|---|---|---|---|---|---|
| | | *Biophysical Model Parameters* | | | |
| P.1 | $v^{(\text{set})}$ | $\text{m}\,\text{d}^{-1}$ | 0.5-1.5 | 0.5 | 1.5 |
| P.2 | $\lambda_O$ | $\text{m}\,\text{d}^{-1}$ | 0.0-0.5 | 0.5 | 0 |
| P.3 | $\lambda_{N(1)}$ | $\text{m}\,\text{d}^{-1}$ | 0.0-0.5 | $8.91\times10^{-2}$ | 0 |
| P.4 | $\lambda_{N(2)}$ | $\text{m}\,\text{d}^{-1}$ | 0.0-0.5 | 0 | 0.5 |
| P.5 | $\kappa_{N(3)}$ | $\text{m}^2\,\text{s}^{-1}$ | 0.0-0.5 | $1.7\times10^{-3}$ | 0 |
| P.6 | $B1$ | - | 10.0-20.0 | 12.5 | 10.0 |
| P.7 | $\varepsilon_{\text{PAR}}$ | - | 0.25-0.75 | 0.38 | 0.61 |
| P.8 | $\lambda_w$ | $\text{m}^{-1}$ | 0.03-0.05 | 0.03 | 0.03 |
| P.9 | $c_P$ | $\text{m}^{-2}\,(\text{mg Chl})^{-1}$ | 0.005-0.045 | 0.05 | 0.005 |
| P.10 | $b_P$ | $\text{d}^{-1}$ | 0.005-0.075 | 0.05 | 0.075 |
| P.11 | $\phi_N^{(\text{opt})}$ | $\text{mmol N}\,(\text{mg C})^{-1}$ | $1.0\times10^{-4}$-$5.0\times10^{-2}$ | $2.62\times10^{-3}$ | $3.6\times10^{-2}$ |
| P.12 | $\phi_P^{(\text{opt})}$ | $\text{mmol P}\,(\text{mg C})^{-1}$ | $1.0\times10^{-4}$-$1.0\times10^{-3}$ | $1.0\times10^{-3}$ | $1.0\times10^{-3}$ |
| P.13 | $\alpha_{\text{chl}}^{(0)}$ | $\text{mg C}\,(\text{mg Chl})^{-1}\mu\text{E}^{-1}\text{m}^2$ | $5.0\times10^{-6}$-$5.0\times10^{-5}$ | $5.0\times10^{-6}$ | $2.9\times10^{-5}$ |
| P.14 | $\theta_{\text{chl}}^{(0)}$ | $\text{mg Chl}\,(\text{mg C})^{-1}$ | 0.005-0.05 | $5.5\times10^{-3}$ | 0.36 |
| P.15 | $r_Z^{(0)}$ | $\text{d}^{-1}$ | 1.0-7.5 | 2.30 | 7.5 |
| P.16 | $d_Z$ | $\text{d}^{-1}$ | 0.025-0.1 | 0.1 | 0.08 |
| P.17 | $\eta_Z$ | - | 0.05-0.55 | 0.55 | 0.55 |
| P.18 | $h_Z^{(F)}$ | $\text{mg C}\,\text{m}^{-3}$ | 50.0-500.0 | 500.0 | 500 |
| | | *Velocity Perturbation Coefficients* | | | |
| C.1 | $C_1^{(1)}$ | - | -10.0(-15.0)-0.0 | -10 | 0 |
| C.2 | $C_2^{(1)}$ | - | -10.0(-20.0)-0.0 | -10 | -20 |
| C.3 | $C_3^{(1)}$ | - | -15.0(-25.0)-0.0 | -15 | -25 |
| C.4 | $C_4^{(1)}$ | - | -20.0-0.0 | -20 | -20 |
| C.5 | $C_5^{(1)}$ | - | -30.0(-15.0)-0.0 | -30 | -15 |
| C.6 | $C_6^{(1)}$ | - | -35.0(-15.0)-0.0 | -35 | -15 |
| C.7 | $C_7^{(1)}$ | - | -50.0(-20.0)-0.0 | -50 | -10 |
| C.8 | $C_8^{(1)}$ | - | -150.0(-30.0)-0.0 | -56 | 0 |
| C.9 | $C_9^{(1)}$ | - | -2895.0(-40.0)-0.0 | -2385 | -18 |
| C.10 | $C_{10}^{(1)}$ | - | -65.0(-30.0)-0.0 | -65 | -30 |
| C.11 | $C_{11}^{(1)}$ | - | -20.0-0.0 | -20 | -15 |
| C.12 | $C_{12}^{(1)}$ | - | -10.0(-15.0)-0.0 | -10 | -15 |
| C.13 | $C_1^{(2)}$ | - | -10.0(-15.0)-0.0 | -10 | -15 |
| C.14 | $C_2^{(2)}$ | - | -10.0(-20.0)-0.0 | -10 | -20 |
| C.15 | $C_3^{(2)}$ | - | -15.0(-25.0)-0.0 | -15. | -25 |
| C.16 | $C_4^{(2)}$ | - | -20.0-0.0 | -20 | -20 |
| C.17 | $C_5^{(2)}$ | - | -30.0(-15.0)-0.0 | -30. | -15 |
| C.18 | $C_6^{(2)}$ | - | -35.0(-15.0)-0.0 | -30.3 | -15 |
| C.19 | $C_7^{(2)}$ | - | -50.0(-20.0)-0.0 | -22. | -20 |
| C.20 | $C_8^{(2)}$ | - | -150.0(-30.0)-0.0 | -78. | -30 |
| C.21 | $C_9^{(2)}$ | - | -2895.0(-40.0)-0.0 | -2010 | 0 |
| C.22 | $C_{10}^{(2)}$ | - | -65.0(-30.0)-0.0 | -65 | -30 |
| C.23 | $C_{11}^{(2)}$ | - | -20.0-0.0 | -20 | -20 |
| C.24 | $C_{12}^{(2)}$ | - | -10.0(-15.0)-0.0 | -10 | 0 |



*Author contributions.* SK and KMS initially implemented the model at BATS. MEM implemented the model at HOTS with contributions from SK. SK, KMS, NP, and PEH developed the model calibration algorithm; SK implemented it. SK performed the calibration in this paper with input from KEN, PN, NSL, and PEH. PN and NSL helped interpret calibration results. The initial draft of this manuscript was prepared by SK and PEH, which was reviewed and edited by MEM, KMS, NP, KEN, and NSL.

*Competing interests.* The contact author has declared that none of the authors has any competing interests.

*Acknowledgements.* SK was supported by the ANSEP Alaska Grown Fellowship and by the National Science Foundation while producing the data for the results presented in this manuscript. This research has also been supported by the National Science Foundation (grant nos. OCE-1924636, OCE-1924658, and NSF 18-573).




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
