# Peer review of "Simultaneous versus sequential estimation of biogeochemical and physical parameters in coupled marine ecosystem models"

_EGUsphere, 2025_

## Author Comment (AC1)

**Review 1:**

*Summary:*

The study advertises automatized parameter optimization in coupled biogeochemical ocean modelling in a 1-dimensional framework. The authors highlight uncertainties in both, physical and biogeochemical model parameters. They obtain best results in fitting these parameters simultaneously when reproducing the mean seasonal cycle of biogeochemical observations at two observing stations (BATS and HOTS).

*Major Comments:*

The study addresses important aspects in biogeochemical ocean modelling and all the work is greatly appreciated. The study is generally well written and organized. Still, I would suggest some clarifications and more discussion on the aspects outlined below in the specific comments. I am particularly puzzled about seemingly strong nutrient budget differences between the different parameter fits at station HOTS, which should be explained in more detail, and I am not 100% sure how sinking of organic matter is treated. Further, the manuscript might benefit from being more concise to enhance readability and attract a wider audience (e.g. it might not be necessary to start with the most general form in the "Optimization Methodology" and all BATS subsections could be merged to avoid repetition (same for HOTS); some parts might well be moved to an Appendix).

> Thank you for taking the time to review our paper. We appreciate the positive comments on the work and the constructive feedback. We have addressed the reviewer's concerns about clarity and length by addressing the points of confusion, looking for opportunities to tighten the content, and rearranging the material. We have addressed the reviewer's feedback point-by-point below.

*Specific comments:*

Ln 6: please add that a 1-dimensional model has been used

> This has been updated to "parameters associated with a one-dimensional physical ocean model."

Ln 9: replace "observational data" by "the observed mean seasonal cycle"

> This has been updated to "the observed mean seasonal cycle."

Ln 11: I find the word "prediction" misleading because the tuned models have not been tested for predicting independent data but the authors rather refer to the mean seasonal cycle used for parameter fitting. Better: "simultaneous estimation results in a closer fit to the observed seasonal cycle of biogeochemical tracers"

> As suggested, the text has been updated to "closer agreement with the mean seasonal cycles for oxygen and particulate organic nitrogen."

Section 1: Introduction

Ln 21ff: How was the conclusion drawn that "sequential parameter tuning in coupled models may not produce accurate predictions"? My understanding so far was that in case the effects of certain parameters compensate it might well make sense to keep some of them on fixed predefined values (cf. Matear, 1995, Löptien & Dietze, 2021). Could this apply here?

> We agree that the issue of parameter compensation can be mitigated if there are a priori biogeochemical or physical justifications for selecting appropriate values of a subset of parameters. In the present case of a 1D physical model coupled to BFM17, specific appropriate values are unknown for nearly all parameter values, and, at most, we are only able to define a range of expected values. Although this limitation may not hold for all current and future model formulations, it is the case in many situations. We have updated the text to clarify this point.

Ln 25: I suggest to reformulate. I don't see that "the potential benefits" are "quantified" – especially in the context of the foregoing sentence - because there is no prediction or test on independent observational data. I would rather say that the goodness of fit of the different approaches is explored with respect to the mean seasonal cycle on two locations.

We agree and have updated this sentence to read: "In this paper, we explore the potential benefits of such an approach by comparing sequential and simultaneous calibrations of a one-dimensional (1D), moderately complex, coupled BGC-physical model based on agreement with the mean seasonal cycle at two different ocean locations."

Ln 26: Add 1-dimensional

This has been updated, as shown in the response above.

Ln 35: Please add that this refers to the mean seasonal cycle.

The text has been updated to "demonstrated good agreement between BFM17+POM1D and the observed mean seasonal cycle from both BATS and the Hawaii Ocean time series."

Ln 42: replace "with the data from BATS and HOTS" by "with the observed mean seasonal cycle of selected biogeochemical tracers at BATS and HOTS"

The text has been updated to "agreed well with the observed mean seasonal cycles from both BATS and HOTS".

Section 2: Optimization Methodology
As I understood, the method depends on some random sampling of the parameters. Would a repetition lead to different results?

The method does include an initial step that randomly samples the parameter space to determine initial parameter values for the subsequent gradient-based optimizations. In this sense, a repeated application of the methodology with different random samples could lead to different results. With enough random samples, however, confidence in the repeatability of the results improves, but this must be balanced against the computational cost of sampling many random points, the majority of which will not be used for subsequent gradient-based optimizations. For example, in the present study we sampled 10,000 initial parameter sets but only used the 20 best performing parameter sets to initialize the gradient-based optimizations. We have added text in Section 2 clarifying this point for the reader.

Ln 66: Why not write right away which objective function has been used instead of providing a general formulation? (if needed you could refer to Kern et al. 2024 for the generalized form)

We agree that the previous methodology section was made overly verbose by reintroducing the general form of the problem instead of relying on the previous paper. We have tightened this section by combining equations and excluding terms from the general formulation that are not relevant in this study. We hope these edits improve readability, reduce confusion, and address the reviewer's concern.

Ln 92: Could you be more specific? "… the initial search is truncated based on the available computational resources."

The number of model evaluations necessary to fully explore the parameter space of a model with 42 parameters is prohibitively expensive to execute. We therefore perform a certain number of evaluations based, in part, on how long it takes to produce that many evaluations. With more computational capacity, the number of evaluations that can be performed increases. This is clarified with updated text at the end of Section 2.

Ln 96/97 Maybe mention already here how Nsamp and Ntop were chosen (or refer to page 19 which feels a bit repetitive)?

We agree and have moved the justification for the selected values to Section 2.

Ln 151: How were "the most sensitive BGC parameters" chosen? (refer to Section 5?)

*A note directing the reader to Appendix A, where the parameter sensitivity section now appears, has been added.*

Section 4: Physical Scenarios

To me the title "Physical scenarios" is somewhat misleading because I (and maybe others) associate "scenarios" with climate change or management scenarios while this just refers to two different locations.

*We have revised and reordered the paper so that there is no longer a section called "Physical Scenarios." We now have separate subsections within the "Results and Discussion" (Section 4) titled "Bermuda Atlantic time-series (BATS)" and "Hawaii Ocean time-series (HOTS)".*

Ln 238: Again, I find "physical scenarios" somewhat misleading here. Better? "locations"

*We have changed the phrase "physical scenarios" everywhere it appears, typically with the word "locations", as suggested by the reviewer.*

Ln 244: better replace "scenario"

*Agreed, see response above.*

Ln 247: better replace "these physical scenarios"

*Agreed, see response above.*

Ln 248: How was the model initialized? Why are sinking velocity and boundary control parameters included as part of the physical model (as stated on page 15, ln 311)? Since the study investigates the separation of physical and biogeochemical parameter tuning, this point deserves attention.

*We thank the reviewer for highlighting this potential point of confusion and now describe the initialization and forcing of the model for the BATS and HOTS locations in the second paragraph of Section 4. Regarding the reasoning for considering the sinking velocity and boundary conditions as part of the physical model in this study, this is mostly a matter of the structure of the governing equation in Eq. (3). The biological terms were considered to be those directly included in the first term of this equation. The sinking velocity is included in the second term to represent the combined vertical transport of particulate organic matter; these combined effects are considered to be part of the physical model. The boundary control parameters were included as part of the physical model because their implementation is dependent on how we are choosing to represent the dynamics. That is, the boundary conditions are not an inherent part of the biological model; rather, their formulation comes from the coupling of the physical and biogeochemical models. We have added text to Sections 3.2 and 3.3 clarifying these points and have also combined all physical parameters into a single table to avoid confusion (see Table 2 in the revised paper).*

Bermuda Atlantic time-series (BATS)

Ln 261: I find "trend" misleading when referring to a seasonal cycle

*We agree and the word "trend" has been updated to "seasonal cycle".*

Ln 263: It would be nice to see some numbers here.

*To make the comparisons more quantitative, we have included the maximum field values for reference in this section.*

Ln 264: I would appreciate a short description of the typical seasonal cycle.

*We agree that this is important and with the reorganization of the paper so that all BATS and HOTS results appear together in Sections 4.1 and 4.2, respectively, it is now easier for the reader to find the descriptions of the typical seasonal cycles for both locations. In particular, both these sections begin with a description of the observed seasonal cycles.*

Ln 265: What is meant by "annual trend"?

> The phrase "annual trend" refers to the observed seasonal cycle in the corresponding quantity. The wording has been updated to clarify our meaning.

Ln 268: Has significance been tested? Otherwise please reword. It would be nice to see some number (e.g. % relative to the observed mean)

> The term was not used in a statistical sense, and the wording has been updated to avoid giving that impression.

Ln 271: How was the model initialized and how come that the nutrient content in the water column is generally over-estimated (is this related to the bottom boundary relaxation?)?

> We now include a description of how the model is initialized in the second paragraph of Section 4. The overestimated nutrients are likely a result of the bottom boundary relaxation coefficients. Much of the initial ad hoc tuning from Smith et al. (2021) focused on tuning the relaxation coefficients and the reviewer is spot on that these are important parameters in determining model accuracy.

Ln 271: I believe "over-predicted" should be "overestimated"

> We agree and the wording has been updated accordingly.

Hawaii Ocean time-series (HOTS)
The nutrient budgets seem visually rather different. Is this related to bottom boundary relaxation? So, is this here defined to be related to the physical model or did I get this wrong?

> The reviewer is correct that the nutrient budget is related to the bottom boundary relaxation and that these parameters are included with the physical model. The additional text at the end of Section 3.3 now provides justification for this choice.

Ln 304: "increase the accuracy of the model" should rather be something like "enhance the fit to the mean seasonal cycle".

> The language has been updated accordingly.

Section 5: Parameter Sensitivity Analysis
The authors might consider to move this part to an Appendix (same for the Twin Experiments).

> We agree that these sections made the original version of the paper quite long and both the parameter sensitivity analysis and twin simulation experiments have been moved to separate appendices.

Ln 311: this did not become clear to me – has "sinking velocity" of organic matter been optimized as part of the physical model? What is the rationale behind this? I would particularly be interested in more details, because foregoing studies considered this parameter to be important (cf. Taucher & Oschlies, 2011 or Kriest & Oschlies, 2008)

> The reviewer is correct that the sinking velocity (i.e., settling velocity) for the particulate organic matter was included in the optimization. It was specifically included in the physical model parameters in Table 2 because of its connection to vertical advection in Eq. (3). We now provide text on this point at the end of Section 3.2.

Ln 318: How were the parameter ranges determined?

> The parameter ranges for biogeochemical parameters were initially taken from supplementary material from the initial description of the full BFM in Vichi et al. (2007). Some of the ranges were updated to be broader during the parameter estimations in Kern et al. (2024). We now include a statement regarding the source of the parameter ranges in Section 3.1.

Ln 332: How much do the base line parameters for BATS and HOTS differ?

> The baseline parameter values are the same for BATS and HOTS, which we now explicitly state in Section 3.1.

Section 7.1: Bermuda Atlantic Time Series
It might be nice to move this part to the first Bermuda Atlantic time-series (BATS)-Section.

> We agree and the paper has been reordered so that the Bermuda Atlantic Time Series discussion is all within Section 4.1. First the observational data is compared to the baseline model implementation, then the parameter estimation results are presented and discussed.

Ln 428: How do temperature and salinity profiles look like compared to the observations?

> The temperature and salinity profiles are directly input to the model from observational data and are therefore "inputs" rather than model "outputs". At the beginning of Section 4 we have attempted to make it clearer that temperature and salinity are inputs rather than state variables that can be compared with the observations.

Section 7.2: - Hawaii Ocean Time Series
Same here - it might be nice to move this part to the first section on HOTS.

> As with the BATS description, the paper has been reordered so that the Hawaii Ocean Time Series discussion is all within Section 4.2. First the observational data is compared to the baseline model implementation, then the parameter estimation results are presented and discussed.

How much do the biogeochemical parameters for BATS and HOTS differ after optimization and what does this mean for global biogeochemical models? (cf. Schartau & Oschlies, 2003)

> This is a very important question and there is indeed a difference in some of the BGC model parameters between BATS and HOTS. This suggests that global BGC models may benefit from allowing for spatial variation in the model behavior across distinct BGC communities. This may also extend to allowing for temporal variability in model parameters. However, this is in part dependent on the model being used. In the present study, we are using a moderately complex model, so our phytoplankton and zooplankton represent community average behavior. With more specificity, the model parameters will theoretically be better defined and require less variability across regions. We have added text to the end of Section 4.2 explaining these points.

Ln 450: How do temperature and salinity profiles look like compared to the observations?

> We have addressed this question above with respect to BATS and have added text in Section 4 to make this point clearer.

Section 8: Conclusions (& Discussion):
The results of the presented study seem somewhat contradictory to earlier findings where it has been shown that biogeochemical model parameters can be tuned to compensate for ocean model differences - which can be problematic when it comes to projections (Löptien & Dietze, 2019; Pasquier et al. 2023). I would be very interested in some thoughts on this. Also, it has been stated early on that the multitude of poorly known biogeochemical model parameters might lead to overfitting (e.g. Matear, 1995). I fully understand that the study/model design and lack of observational data makes testing of the presented models on independent observations difficult - still some discussion on overfitting and strategies for (future) model testing would be beneficial. Finally, it should be mentioned that the obtained results might well depend on parameter choice, location and objective function. It is particularly worth mentioning that even the physical model parameters were fitted on selected biogeochemical observations while temperature and salinity (as I understood it) were not considered.

> We agree that the results will depend on a number of different factors. including all those mentioned by the reviewer. To make these point more explicit for the reader, we have included a statement in the final paragraph of the conclusions explaining that the results are dependent on the parameter choices, study locations, and formulation of the objective function. On the issue of overfitting, we agree that this is a concern, and we have similarly added text to the conclusions providing possible approaches to mitigating this issue.

Ln 466: remove "scenario"

> This has been changed to "location".

Ln 470: I find it confusing to talk about "trends" when it comes to the seasonal cycle

> We agree and we have changed this to "seasonal cycle".

*References:*
Kern, S., McGuinn, M. E., Smith, K. M., Pinardi, N., Niemeyer, K. E., Lovenduski, N. S., & Hamlington, P. E. (2024). Computationally efficient parameter estimation for high-dimensional ocean biogeochemical models. Geoscientific Model Development Discussions, 2023, 1-34.

Kriest, I., & Oschlies, A. (2008). On the treatment of particulate organic matter sinking in large-scale models of marine biogeochemical cycles. Biogeosciences, 5(1), 55-72.

Löptien, U., & Dietze, H. (2017). Effects of parameter indeterminacy in pelagic biogeochemical modules of Earth System Models on projections into a warming future: The scale of the problem. Global Biogeochemical Cycles, 31(7), 1155-1172.

Löptien, U., & Dietze, H. (2019). Reciprocal bias compensation and ensuing uncertainties in model-based climate projections: pelagic biogeochemistry versus ocean mixing. Biogeosciences, 16(9), 1865-1881.

Matear, R. J. (1995). Parameter optimization and analysis of ecosystem models using simulated annealing: A case study at Station P.

Pasquier, B., Holzer, M., Chamberlain, M. A., Matear, R. J., Bindoff, N. L., & Primeau, F. W. (2023). Optimal parameters for the ocean's nutrient, carbon, and oxygen cycles compensate for circulation biases but replumb the biological pump. Biogeosciences, 20(14), 2985-3009.

Schartau, M., & Oschlies, A. (2003). Simultaneous data-based optimization of a 1D-ecosystem model at three locations in the North Atlantic: Part II—Standing stocks and nitrogen fluxes.

Taucher, J., & Oschlies, A. (2011). Can we predict the direction of marine primary production change under global warming?. Geophysical Research Letters, 38(2).

**Review 2**

*Review:*
This study investigates the optimisation of physical and biogeochemical parameters independently, sequentially, and simultaneously and concludes that the combined optimisation approach performs best. The topic is important and relevant, and the results could make a valuable contribution to the field. However, to strengthen the paper and make the conclusions more convincing, I suggest the following improvements.

> Thank you for taking the time to review our paper. We appreciate the positive comments on our work and on the contribution it will make to the field. We especially appreciate the constructive feedback and have tried to address all reviewer points to improve the impact of this work.

*Major Comments:*
1. Demonstrate simultaneous optimisation in the twin experiment

The paper's central claim is that simultaneously optimising physical and biogeochemical parameters provides the best outcome. To clearly demonstrate this, I recommend expanding the twin experiment to explicitly include this combined optimisation case. Showing results in an idealised setting would help clarify both the advantages and potential challenges of the approach, thereby making the BATS and HOTS applications more robust and convincing.

> We appreciate the feedback from the reviewer on this point. There are indeed many ways the twin simulation experiments (TSEs) can be used and in the present study we focused specifically on the verification of the parameter estimation methodology itself, including ensuring proper coupling between BFM17+POM and DAKOTA. For this purpose, expanding the TSEs to also include physical parameters would only have served to demonstrate that the methodology can be applied to an even broader set of parameters. Although this study could still be valuable, to avoid distracting from the primary focus of this paper – namely, determining whether simultaneous rather than sequential parameter estimation gives better agreement with observational data – and to address a suggestion from the other reviewer, we have moved the TSE section to an appendix and we now mention the purpose of the TSEs and the possibility for expanded TSE studies at the end of Section 3.1.

2. Investigate parameter correlations and identifiability

A more thorough analysis of parameter correlations would significantly strengthen the paper. Earlier work (e.g. Matear 1995) showed that many biogeochemical model parameters are highly correlated and thus cannot be independently determined. Is this also the case for your BFM17 model? Figure 6 shows that two biogeochemical parameters are correlated and not uniquely determined. Related to point (1), simultaneously optimising physical and biogeochemical parameters may amplify these correlations. Quantifying and discussing these relationships would provide valuable insight into parameter uniqueness and the robustness of the optimisation results.

> We agree with the reviewer that correlations can be important and are often an opportunity to reduce the cost of the optimization. Although the present methodology, which follows the approach outlined and validated in Kern et al. (2024), does not explore or take advantage of correlations, we have mentioned the importance of pursuing this topic in the final paragraph of the conclusions.

3. Address potential overfitting and define acceptable-fit criteria

For real observational data, the goal should not be a perfect fit to the data, as the model is an approximation of reality. The optimisation framework should therefore incorporate a notion of what constitutes an "acceptable" fit to the data. Overfitting could lead to unrealistic model behaviour when small perturbations occur (e.g., minor changes in mesoscale upwelling). Defining an appropriate tolerance level or misfit threshold would make the results more physically and statistically robust.

> This is an important point, and we agree that overfitting is a concern when using optimization techniques to estimate model parameters. In the revised paper we now bring more attention to this point by including a statement in the introduction acknowledging this concern and in the conclusions we now mention potential ways that overfitting may be addressed – including regularization and Bayesian methods – using the present approach as a framework.

4. Clarify and evaluate the mesoscale upwelling parameterisation

The physical model uses an idealised parameterisation of mesoscale upwelling and downwelling. Please provide evidence, references, or justification for representing mesoscale variability in this way. It would also strengthen the paper to assess whether the optimised results are consistent with known or observed seasonal variations in mesoscale activity. This evaluation would help determine whether the model's physical representation is realistic and whether the optimised parameters remain physically meaningful.

> We appreciate the reviewer for pointing out this area of improvement. Immediately before Eq. (4) we now provide additional references explaining the origin of the parameterizations used for the vertical advection. These parameterizations go back to the original BFM description in Bianchi et al. (2006) and were also used in the BFM17 studies by Smith et al. (2021) and Kern et al. (2024). At the end of Section 4.1 we highlight that for the BATS location, the seasonality of the parameterization does follow observed trends in the mesoscale activity. In particular, although the transport specifically related to mesoscale eddies alone has not been previously quantified for direct comparison to our results, we do have evidence from Mahadevan & Archer (2000) and Salmon et al. (2015) that there should be increased vertical transport from mesoscale eddies. We also know from Aguedjou et al. (2019) that there should be more mesoscale eddies in the fall and winter. Unfortunately, we were not able to find similar (even indirect) information on mesoscale eddy transport at HOTS and now explicitly note this in the text near the end of Section 4.2 as an area for future study.

*Specific Comments:*

Line 70: Please clarify what is considered an acceptable fit to the data. Does a normalised RMS error of 1 define this threshold?

> We thank the reviewer for identifying this potential source of confusion. In general, the smaller the normalized error the better, and the most significant error value is 0, corresponding to identical observational and model fields. However, values of the normalized RMSD close to or less than 1 indicate that the variability between the observational and model results is below the variations in the observations, suggesting a good model fit. Through the simultaneous parameter estimations, we have been able to produce normalized error values below 1 for all fields other than oxygen. We now explain these points in the revised paper after Eq. (2).

Model constraints: Are any constraints applied to the model parameters beyond the prescribed range limits?

> There are no constraints applied to the model parameters other than the min and max values in Tables 1 and 2.

Cycle enforcement: Is an annual repeating cycle imposed on the simulated fields? If so, how is this implemented?

> There is a repeated annual cycle imposed via the input temperature, salinity, and vertical velocity fields. The BGC fields are allowed to adjust accordingly. To address this point, we have now made the forcing procedure clearer at the beginning of Section 4 in the revised paper.

Line 115: It is unsurprising that the conjugate gradient approach fails here; stochastic methods such as simulated annealing are typically more robust in complex optimisation problems.

> We agree and we tested algorithms including conjugate gradient, COBYLA, and quasi-newton (QN), all of which are available in DAKOTA. Of those tested, QN performed the best, although

simulated annealing and genetic algorithms are worth exploring in future work. We now note the importance of exploring other optimization methods at the end of the Conclusions.

Line 155: Is the settling velocity depth-dependent? Please clarify.

The settling velocity is not depth dependent and is constant for a given BGC state variable. It is set to zero for variables not associated with the organic detritus. This has now been clarified at the end of Section 3.2.

Line 165: Add clarification that at both sites, the Ekman velocities are all negative.

The corresponding line has been updated to read: "The resulting profiles of $W$ for BATS and HOTS are all negative, resulting in purely downward advection."

Line 169: Do you have evidence or references supporting the chosen parameterisation of eddy vertical velocities?

The parameterization of the eddy velocity is based on the parameterization of the general circulation velocity which uses the Ekman pumping as a grounding. The approach used here follows the history of the model development. It was used in Smith et al.'s (2021) formulation of the BFM17+POM1D model, which was based on the work by Bianchi et al. (2006). We have added text immediately prior to Eq. (4) clarifying these points for the reader.

Table 1: The September values are notably large—can you provide a physical explanation for this?

This is an astute observation by the reviewer and was due to the random nature of the baseline values from Smith et al. (2021). We now point out in the description of Table 2 that the baseline values have been randomly generated. We also note that the maximum possible eddy velocity was 0.2 md$^{-1}$. As a result, because the maximum general circulation velocity for September was low, the possible coefficients are much bigger than the other months.

Line 235: What is the model time step used in the simulations?

We have added text at the beginning of Section 4 outlining how the model was run, including the time step. Specifically, the time step was 400 seconds, although data was output as daily averages.

Figure 6: If two parameters are highly correlated, why attempt to optimise both? Consider removing one and instead testing the 13th parameter previously identified as significant.

The reviewer raises an important point, and we note in Appendix A (where the sensitivity analysis now appears) that 3 of the 13 most sensitive parameters are from the same equation in BFM17 [i.e., Eq. (A34) from Smith et al. (2021)]. Although the three parameters are not directly correlated due to the complexity of the equation in which they appear, we did find that the TSE was most successful when including only two of the three parameters. This is now explained in more detail in Appendix B.

Line 370: Why not use twin experiments to directly compare independent, sequential, and simultaneous optimisation performance?

The reviewer has identified an important point about our overall approach used to compare sequential and simultaneous optimization approaches. The most important reason for our approach is that we need improved model implementations. The BATS and HOTS model implementations, at baseline, are tuned in an ad hoc manner for the BATS site. Therefore, they do not represent a 'true' case, which would be needed for a 'target' data set in thorough twin experiments. So, this work was done to get a better prediction of the best model implementation for each site. We tested two paths which were worth considering and compare them using normalized RMSD values. It would be worthwhile in future work to use the better implementations as target cases in twin experiments, but we required the better model implementations

themselves. As we now note at the end of Section 3.1 in the revised paper, the limited twin simulation experiments included in the paper primarily demonstrate that the optimization algorithm is correctly implemented and sufficient for taking on the problem we are applying it to.

Table 4: What constitutes an acceptable fit (e.g., RMS = 1)? What would overfitting look like in this context?

As noted in our response to the reviewer's comment on Line 70, we did not implement a threshold for what would be considered an acceptable normalized RMSE value. The intent was to see how far each method minimized the error.

Figure 7: Does the seasonal behaviour of W_e make physical sense? For example, does eddy activity exhibit seasonal variation? The minimum in September (Fig. 7c) coincides with a large value in Table 1—please comment. Similar behaviour appears in Fig. 8c; again, some discussion would be useful.

From our understanding, the seasonal behavior of $W_e$ does make sense, but there are no existing direct observations of these quantities at either the BATS or HOTS locations. Mahadevan & Archer (2000) looked at simulation results of a high-resolution model at BATS and HOTS, finding that there is more vertical transport associated with mesoscale eddies which are not represented in lower resolution models. They suggest that the mesoscale and even frontal scale may need to be resolved for accurate predictions of the biogeochemistry. They help us establish that we expect higher transport rates when there are more active mesoscale eddies, but they do not tell us anything about the annual cycle in eddy velocities. We now discuss these points at the end of Sections 4.1 and 4.2 in the revised paper.

*Reference:*
Matear, R.J. (1995). Parameter optimization and analysis of ecosystem models using simulated annealing: A case study at Station P. Journal of Marine Research, 53, 571–607.